# Evidence for millennial-scale interactions between Hg cycling and hydroclimate from Lake Bosumtwi, Ghana

Alice R. Paine[1,2]*, Joost Frieling[1], Timothy M. Shanahan[3], Tamsin A. Mather[1], Nicholas McKay[4], Stuart A. Robinson[1], David M. Pyle[1], Isabel M. Fendley[1,5]*, Ruth Kiely[6], William D. Gosling[6]

[1]Department of Earth Sciences, University of Oxford, UK, OX1 3AN
[2]*Department of Environmental Sciences, University of Basel, Bernoullistrasse 32, CH-4056 Basel, Switzerland
[3]Department of Earth and Planetary Sciences, University of Texas at Austin, Texas, USA
[4]School of Earth and Sustainability, Northern Arizona University, Flagstaff, Arizona, USA
[5]Department of Geosciences, Pennsylvania State University, University Park, PA, USA
[6]Institute for Biodiversity & Ecosystem Dynamics, University of Amsterdam, Amsterdam, Netherlands

**Corresponding Author**: alice.paine@unibas.ch
*current affiliation*

**Changing hydrology impacts the biogeochemical cycling of elements such as mercury (Hg), whose transport and transformation in the environment appear linked to hydroclimate on diverse timescales. Key questions remain about how these processes manifest over different timescales and their potential environmental consequences. For example, millennial-scale Hg-hydroclimate interactions in the terrestrial realm are poorly understood, as few sedimentary records have sufficient length and/or resolution to record abrupt and long-lasting changes in Hg cycling, and the relative roles of depositional processes on these changes. Here, we present a high-resolution sedimentary Hg record from tropical Lake Bosumtwi (Ghana, West Africa) since ~96 ka. A coupled response is observed between Hg flux and shifts in sediment composition, the latter reflecting changes in lake level. Specifically, we find that the amplitude and frequency of Hg peaks increase as the lake level rises, suggesting that Hg burial was enhanced in response to an insolation-driven increase in precipitation at ~73 ka. A more transient, threefold increase in Hg concentration and accumulation rate is also recorded between ~13 and 4 ka, coinciding with a period of distinctly higher rainfall across North Africa known as the African Humid Period. Two mechanisms, likely working in tandem, could explain this correspondence: (1) an increase in wet deposition of Hg by precipitation and (2) efficient sequestration of organic-hosted Hg. Taken together, our results reaffirm that changes in hydroclimate, directly and/or indirectly, can be linked to millennial-scale changes in tropical Hg cycling, and that these signals can be recorded in lake sediments**.

KEYWORDS: lacustrine, geochemistry, sediment, sapropel, organic, Africa

## 1. Introduction

Mercury (Hg) is a volatile and toxic metal released into the atmosphere as a result of natural
processes (e.g., volcanism, geothermal activity, weathering; Edwards et al., 2021; Selin, 2009) and,
more recently, human activities (e.g., industrial activities, mining, coal burning; Amos et al., 2015).
Approximately 95 % of atmospheric Hg exists in gaseous elemental form ($Hg^0$). An atmospheric
lifetime of up to 2 years permits its transport over long distances prior to removal by wet or dry
deposition (Lyman et al., 2020). Once free gaseous ($Hg^0$) and/or oxidised ($Hg^{II}$) Hg has been
deposited into the terrestrial environment, two sets of reactions become particularly important: (1)
oxidation-reduction, and (2) methylation-demethylation (Branfireun et al., 2020). The reduction of $Hg^{II}$
to $Hg^0$ can result in release back into the atmosphere. $Hg^{II}$ can also be bound to organic matter (OM)
or sulphides, and thus be sequestered and accumulated in sediments (Åkerblom et al., 2013; Hsu-
Kim et al., 2013; Mason et al., 2000). Accumulation of Hg in the terrestrial environment is therefore a
function of the balance between Hg removal from and re-emission to the atmosphere, and governed
by the rate and intensity of different thermal, photo, and biogenic reactions (Bishop et al., 2020; Obrist
et al., 2018).
The exchange of Hg between the terrestrial biosphere, hydrosphere, critical zone, and atmosphere
are intrinsically coupled to climate. Changes in ecosystem Hg loading, overland transport, and
methylation have all been directly linked to decadal-scale changes in global temperature and
precipitation, and their associated shifts in terrestrial productivity, land-atmosphere exchange, and
wildfire dynamics (Bishop et al., 2020; Li et al., 2020). However, studying the long-term natural Hg
cycle presents several challenges. For example, the overwhelming increase in anthropogenic Hg
fluxes in recent decades have substantially altered the environmental dynamics of this cycle,
complicating assessment of how long-term climate change may alter its rate, intensity, and evolution
(United Nations Environment Programme, 2018). Pre-industrial-age archives allow for clear
comparison between natural and anthropogenic-driven changes in Hg cycling, and identification of
regions that may be most vulnerable to the negative effects of these changes, and highlight the
importance of understanding the long-term Hg cycle  (e.g., Cooke et al., 2020; Segato et al., 2023).

**1.1. Mercury cycling and hydroclimate.**
The transport and transformation of Hg at the Earth's surface is linked to the hydrological cycle
(Bishop et al., 2020; Selin, 2009). Water plays a direct role in the efficiency of both Hg deposition and
re-emission. For example, changes in precipitation amount can influence the proportion of Hg
removed from the atmosphere by wet versus dry deposition, with higher precipitation amounts
generally corresponding to enhanced Hg deposition at the surface (Amos et al., 2015; Guédron et al.,
2018). Elevated Hg concentrations have been measured in equatorial ocean surface waters
corresponding to inter-annual peaks in precipitation, with general circulation model simulations
suggesting that these are likely due to higher net Hg flux by wet deposition (Kuss et al., 2011;
Soerensen et al., 2014; Sprovieri et al., 2010). Multi-year monitoring by the Global Mercury
Observation System (GMOS) has similarly revealed distinct interannual differences in total wet

deposition of Hg, with the highest fluxes typically occurring in the wettest years (Leiva González et al., 2022; Sprovieri et al., 2017). Precipitation also facilitates Hg transport in terrestrial watersheds, with simultaneous increases in river discharge, surface runoff, and soil erosion during and after intense storm events all enhancing hydrological 'connectivity' between surface environments and feeder tributaries (Bishop et al., 2020). This enhances overland transport of Hg and other suspended materials, and subsequently their delivery to lake and near-shore marine sediments (Liu et al., 2021; Zaferani and Biester, 2021).

Water also plays an indirect role in drawdown and sequestration of Hg in aquatic environments. Systems that are particularly sensitive to changes in water balance (e.g., terrestrial lakes) are most likely to experience distinct, hydro-climate driven environmental changes that impact their internal Hg cycle (Branfireun et al., 2020). For example, changes in organic matter cycling between the catchment and the lake (Ravichandran, 2004), algal scavenging (Outridge et al., 2019), and early diagenesis (Frieling et al., 2023). A decline in the total water volume of a basin may also elicit a reduction in stratification (Woolway et al., 2020), where increased mixing would ventilate bottom waters and reduce organic-matter burial (Gulati et al., 2017). Conversely, a simultaneous increase in total water volume and nutrient influx may increase stratification and bottom-water anoxia to such an extent, that the system experiences a distinct increase in organic matter burial (Pilla et al., 2020). Studies have also found catchment and basin structure to be important when considering the extent to which sedimentary Hg signals reflect hydroclimate-driven variability, as both influence how easily water is able to transport Hg to, from, and between discrete terrestrial sinks (Paine et al., 2024).

In the short-term, variability in hydroclimate may manifest as annual changes in rainfall intensity and seasonality, or by sub-decadal fluctuations in regional-scale climate modes (e.g., El-Nino Southern Oscillation, North Atlantic Oscillation; Hernández et al., 2020). In the long-term, variability in the form of prolonged droughts and/or pluvials may occur in response to global-scale atmospheric reorganization lasting centuries, and changes in the planet's orbital configuration on timescales of many millennia (Bradley and Diaz, 2021). These wet-dry oscillations are important on a continental-scale. For example, periods of extreme hydroclimate variability are known to have caused major changes in environmental conditions across sub-Saharan Africa during the late Pleistocene, lasting for multiple millennia (e.g., Scholz et al., 2007).

Millennial-scale changes in hydroclimate may also affect the Hg cycle. Sediment cores extracted from the Pacific and Atlantic oceans show low-amplitude Hg signals corresponding to orbital-scale ($>10^4$-year) changes in precipitation and rates of sediment delivery to the ocean (e.g., Chede et al., 2022; Fadina et al., 2019; Figueiredo et al., 2022; Zou et al., 2021), and a growing number of terrestrial successions also show Hg fluctuations coeval with climate-driven changes in local precipitation, cloud formation, and ice/permafrost extent (e.g., Guédron et al., 2018; Nalbant et al., 2023; Paine et al., 2024; Pan et al., 2020; Pérez-Rodríguez et al., 2018). However, few terrestrial Hg records extend beyond the present interglacial (>12 ka), and even fewer come from the low-latitudes, where tropical rainforest, grassland and desert biomes are highly sensitive to millennial-scale hydroclimate variability (Bradley and Diaz, 2021; Schneider et al., 2023). Thus, our current understanding of Hg behaviour

may not fully account for the impact of major, long-term hydroclimate changes on Hg transformation
and transport through tropical environments (Obrist et al., 2018; Schneider et al., 2023), highlighting
the need for new Hg records spanning long (>10$^3$-year) timescales.

## 1.2. Research objectives
Sedimentary records offer an opportunity to assess the impact of millennial-scale hydroclimate
variability, and related effects, on the terrestrial Hg cycle. In sub-Saharan Africa, the West African
Monsoon (WAM) regulates precipitation amount and distribution, and drives long-term evolution of
environmental characteristics and mineral-dust emissions (Kaboth-Bahr et al., 2021; Kuechler et al.,
2013; O'Mara et al., 2022; Weldeab et al., 2007). Proxy records from this domain show that orbitally-
driven variations in the strength of the WAM have frequently driven distinct arid (Cohen et al., 2007;
Scholz et al., 2007) and humid periods (Armstrong et al., 2023; Menviel et al., 2021) throughout the
Pleistocene. These humid and arid periods have been linked to distinct changes in vegetation
structure, ecosystem dynamics, and human evolution across the continent (e.g., Cohen et al., 2022;
Foerster et al., 2022; Gosling et al., 2022b). In light of growing evidence for a hydroclimatic influence
on the terrestrial Hg cycle (e.g., Guédron et al., 2018; Nalbant et al., 2023; Paine et al., 2024), we
hypothesized that humid and/or arid periods in sub-Saharan Africa would have elicited measurable
changes in the Hg cycle, producing measurable signals in regional sedimentary records. Here our
focus is on sediment core BOS04-5B extracted from Lake Bosumtwi, Ghana (West Africa): a core that
provides a clear and continuous record of this hydroclimate variability covering the late Pleistocene
(Koeberl et al., 2007).
Lake Bosumtwi is a closed system isolated from the regional groundwater network, rendering it
sensitive to both short- and long-term variability in rainfall, humidity, and dynamic surface processes
(Shanahan et al., 2008b; Turner et al., 1996). Integrated proxy data shows that Lake Bosumtwi
experienced dramatic changes in water balance, aeolian dust inputs, and biological productivity
throughout its history. These changes all correspond to moisture-driven oscillations between a forest
and grass-dominated catchment in response to insolation-driven variability in WAM strength, and
migration of the Intertropical Convergence Zone (ITCZ) (Gosling et al., 2022a; Miller et al., 2016; Peck
et al., 2004; Vinnepand et al., 2024). Focussing on the uppermost ~47 m of the Lake Bosumtwi
sediment record, this study assesses whether major shifts in local hydroclimate produced measurable
changes in how Hg has been transported to, and buried within, this system since ~96 ka. By
comparing our sedimentary Hg record with proxy data from archives across the African continent
(e.g., Foerster et al., 2022; Scholz et al., 2007), we explored whether hydroclimate has exerted a
measurable effect on terrestrial Hg cycling in the WAM domain in over the past ~100-kyr.

# 2.  Site Description

## 2.1. Lake Bosumtwi

Lake Bosumtwi is the only natural lake in Ghana, West Africa (6°30' N, 1°25' W) (**Fig. 1**). It occupies a meteorite impact crater dated to 1.08 ± 0.04 Ma, which is one of the youngest and best preserved impact craters on Earth (Jourdan et al., 2009). The surrounding bedrock and meteorite impact target rocks are Proterozoic metagraywackes, phyllites and metavolcanic rocks of the Birimian Supergroup (~2 Ga) (Jones et al., 1981). Lake beds, soils, and breccias constitute the most recent rock formations at the site, and are associated with evolution of the crater through time (Koerberl et al. 2007). The present-day lake is ~8.5 km in diameter with a water depth of up to 80 m at the centre, and the current water level is at least 120 m below the crater rim (Shanahan et al., 2007). The crater itself is ~10.5 km in diameter at the rim with steep slopes, and a well-defined spillway (~120 m above the present lake surface) marks evidence of lake overflow likely during the most recent humid period (**Fig. 1**) (Shanahan et al., 2015). The lake is meromictic, with a shallow oxycline located ~10–15 m below the water's surface (Turner et al., 1996).

The Bosumtwi basin is hydrologically closed with no external drainages, connection to the regional groundwater aquifer, river or stream inflow originating outside of the crater (**Fig. 1**) (Turner et al., 1996). Only during exceptionally high lake levels does water leave the lake, through the spillway (Shanahan et al., 2007). Approximately 300 m of sediment has accumulated in the centre of the basin originating from biological processes within the lake, progressive erosion of the crater wall, aeolian transport, and vegetation within the crater (Koeberl et al., 2005). These properties render the lake highly sensitive to changes in atmospheric processes, but also imply that Hg inputs may originate exclusively from the atmosphere (e.g., by wet deposition). Therefore, this system is ideally suited for exploring whether specific basin characteristics (e.g., depth, nutrient status, bathymetry) could also measurably affect how Hg signals are encoded in the sedimentary record.

### 2.1.1. West African Climate

Seasonal variability in the tropical rain belt position drives short-term hydroclimate change in West Africa. During boreal summer, an increase in northern hemisphere summer (June to August) insolation triggers a northward ITCZ shift, creating a pressure gradient that brings moisture eastwards from the Atlantic Ocean to western Africa. The opposite occurs in boreal winter (December to February), where the ITCZ is displaced southwards, bringing dry, aerosol-rich, continental trade winds to West Africa. Together, these seasonal shifts produce distinct annual wet (May to October) and dry seasons.

On longer (>$10^4$-year) timescales, hydroclimate variability in West Africa has been linked to cyclic changes in Earth's orbital configuration. Changes in axial precession produce fluctuations in seasonal insolation above the African continent, influencing the strength of the WAM, the Walker Circulation, the position and dimensions of the ITCZ, and the availability of continental moisture (Gosling et al., 2022b; Kaboth-Bahr et al., 2021; Trauth et al., 2021). Several studies have shown weakening of the

WAM and southward migration of the ITCZ in response to high precession and/or changes in insolation gradient, producing drier conditions in West Africa and subsequent reductions in terrestrial precipitation, ecosystem productivity, and recession of terrestrial forests . Conversely, strengthening of the WAM and a more northerly ITCZ position is documented when precession is low, bringing wetter and warmer conditions to West Africa and causing expansion of dense forests, voluminous lakes, and diverse ecosystems (Larrasoaña et al., 2013; Pausata et al., 2020). During the last glacial cycle, moisture availability in West Africa also fluctuated in conjunction with the waxing and waning of high-latitude ice sheets, and their effects on sea-surface temperatures (SSTs) in the North Atlantic (deMenocal, 1995; Weldeab et al., 2007). This teleconnection exists as a function of atmospheric moisture transport and convection processes occurring in the polar and tropical regions. Low global ice volumes typically correspond to warmer North Atlantic SSTs, driving increased atmospheric moisture transport and hence more moist conditions in West Africa. Conversely, high global ice volumes generally correspond to cooler SSTs in the North Atlantic, and subsequently drier conditions in West Africa (e.g., Crocker et al., 2022; Lupien et al., 2023; Stager et al., 2011; Tjallingii et al., 2008).

### 2.1.2. Local hydrology and vegetation

Approximately 49% of the Bosumtwi drainage basin (area: 106 km$^2$) is currently occupied by the lake. Situated in close proximity to the (current) ecological transition-zone between savannah in the north and moist forest in the south, Lake Bosumtwi lies directly in the seasonal migration path of the ITCZ (Nicholson, 2013). It experiences a current mean annual temperature of ~26$^o$C, ranging between ~23$^o$C in August to ~27$^o$C in February, with cooler temperatures attributed to increased cloudiness and related reduction in incoming solar radiation (Shanahan et al., 2007). Present-day humidity ranges from ~85% in August to ~75% in January, and average annual precipitation is ~1450 mm (Turner et al., 1996). At present, the surrounding catchment as classified as a '*Tropical and Subtropical Moist Broadleaf Forest'* biome (White, 1983), meaning it is heavily forested with well-developed tropical soils, although many flat-lying areas have been converted to agriculture (e.g., maize, plantain, cocoa, and oil palm) in recent decades (Boamah and Koeberl, 2007). Before human occupation of the site, the lake was surrounded by a moist semi deciduous forest, with a canopy including an abundance of trees from the Ulmaceae and Sterculiaceae (flowering plant) families (Miller and Gosling, 2014).

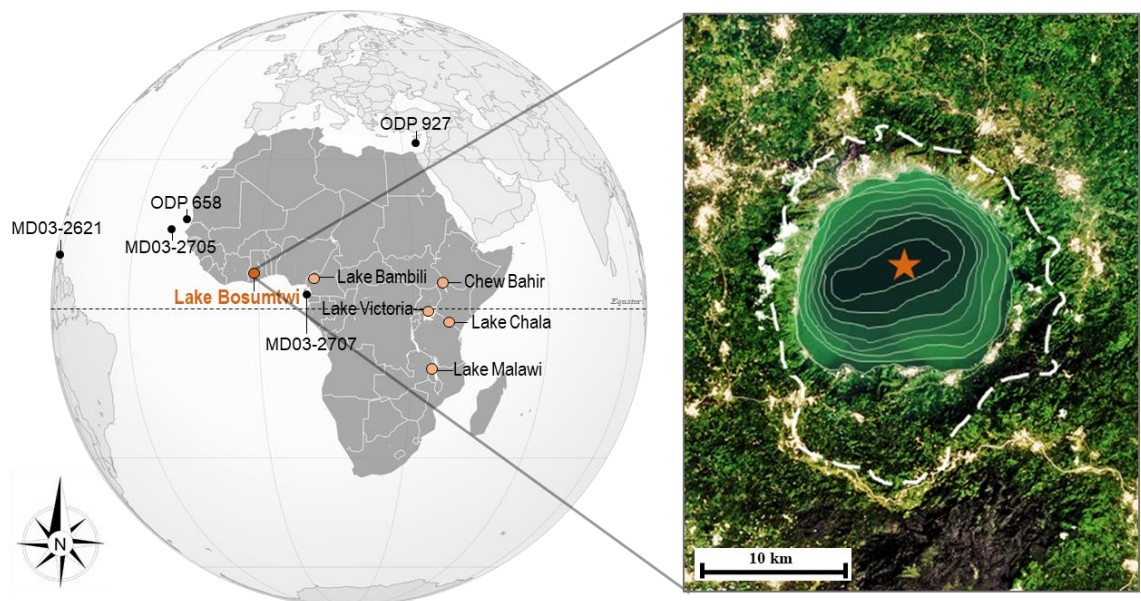

**Figure 1:** (Left) Location of key lake and marine sediment archives in and around Sub-Saharan Africa (SSA). (Right) An aerial photograph of Lake Bosumtwi (*copyright*: NASA, 2018), with the location of the BOS04-5B drill site marked as an orange star. Contours show lake bathymetry, with each step representing a 10 m depth change and culminating in a maximum depth of 75 m (Shanahan et al., 2012). The spillway notch is in the eastern rim of the crater, and the dashed white line marks the extent of the drainage divide (Brooks et al., 2005; Shanahan et al., 2006).

### 2.1.3. Paleoclimatic significance

Lake Bosumtwi provides an excellent record of millennial-scale hydroclimate change in West Africa. The upper ~47 m of sediment corresponds to the interval ~96 ka to present, and contains a series of distinct lithological features suggesting pronounced, climate-driven changes in lake level, catchment structure, and sediment transport processes (Vinnepand et al. 2024; Gosling et al., 2022a; McKay, 2012; Miller et al., 2016). For example, a massive, clastic-rich blue-grey clay unit is present between ~34 and 32 m depth, where TOC values drop to <1% and bulk density values increase by >60% (Scholz et al., 2007). Interpreted and referred to as Arid Interval(AI)-1 (McKay, 2012), this unit formed during extremely dry climatic conditions leading to near-total desiccation of the lake, and this interpretation is further supported by identification of a clear erosional unconformity corresponding to the age of the AI-1 unit in seismic reflection profiles (Brooks et al., 2005; Scholz et al., 2007).

Changing physical properties and geochemistry of sediments deposited prior to and following Unit AI-1 appear to reflect regional hydroclimate shifts (Scholz et al., 2007). Prior to AI-1, the presence of clastic-rich, organic-depleted sediments suggest a progressive reduction in water depth, likely in response to a (long-term) negative water balance (McKay, 2012; Shanahan et al., 2008b). Following AI-1, an abrupt reduction in clastic material concentrations, increased TOC, diagenetic carbonate, and lamination frequency all imply oxygen depletion at the sediment-water interface – evidence for a shift to a positive water balance (McKay, 2012; Scholz et al., 2007; Shanahan et al., 2008a). The core also shows distinct co-enrichment in manganese (Mn) and iron (Fe) in certain intervals following AI-1, that

are associated with manganosiderite (Mn-rich $FeCO_3$) precipitation in the lake sediments.
Manganosiderite requires anoxic non-sulphidic (ferruginous) pore-water conditions and high dissolved
inorganic carbon concentrations to precipitate (Brumsack, 2006; Tribovillard et al., 2006), and
appears as a consequence of the redox tower migrating into the water column: a response to
increasing water column stratification, and overall lake level (Shanahan et al., 2006). The closed
hydrology of Lake Bosumtwi means that changing water levels will primarily reflect the magnitude of
precipitation variability in the region, with higher lake levels typically occurring during wetter climate
intervals (Russell et al., 2003; Shanahan et al., 2008b). However, lake level may have also been
influenced by secondary processes such as evaporation, and, over long time-scales, sediment infill
(McKay, 2012)

## 3. Material and methods

### 3.1.  BOS04-5B

Core BOS04-5B was recovered in 2004 as part of the International Continental Drilling Program
(Koeberl et al., 2005) (full details in **ST1**). Our study focuses on the upper ~47 m section of a 296-m-
long core extracted from deep-water (76 m) site 5 (core BOS04-5B; **Fig. 1**), that extends from the
present-day lake floor to the brecciated bedrock dated by $^{40}Ar/^{39}Ar$ to 1.08±0.04 Ma (Jourdan et al.,
2009). Over ~67% of the full 294 m-long (~1-Myr) sediment succession is laminated (Koeberl et al.,
2007), with distinctive alternating clastic, organic and carbonate laminae (Shanahan et al., 2012,
2009, 2008a). Thicker laminations are visible either as packets of light grey microturbidites, or distinct
yellow and orange carbonates produced by enhanced redox-related precipitation of Fe and Mn
bearing minerals (Shanahan et al., 2008a).
After drilling in 2004, core BOS04-5B was shipped to the University of Rhode Island and split. The
physical properties of the full ~296 m core were measured at 2-cm intervals using a Geotek ® multi-
sensor core logger (Koeberl et al., 2007). After logging and imaging and at 4-cm resolution, 2 cm thick
slices were removed from the core half and separated into sub-samples for multi-proxy analyses,
including sediment magnetic hysteresis, x-ray diffraction mineralogy, total organic and inorganic
carbon content, bulk organic carbon and nitrogen isotopes, grain size, pollen, and charcoal (e.g.,
Gosling et al., 2022; McKay, 2012; Miller et al., 2016). Following bulk sediment analyses, the BOS04-
5B core material was transferred to the Continental Scientific Drilling (CSD) Repository in
Minneapolis.

### 3.2.   Chronology

Age control for the ~47 m of sediment analysed in this study is provided by the BOSMORE7 model,
presented by Gosling et al. (2022). Using a combination of radiocarbon (calibrated $^{14}C$; n= 109),
optically stimulated luminescence (OSL; n=22) and uranium-thorium (U/Th; n=5) dates as
independent tie-points, Bayesian modelling suggests that the upper ~47 m of sedimentation at Lake
Bosumtwi corresponds to the interval ~96–0 ka (full details in **ST2**) (Gosling et al., 2022a; Shanahan
et al., 2013). Age estimates for the AI-1 sedimentary unit are constrained by [14]C, OSL and U-Th
dating of the surrounding sediments, and suggest this unit formed between 77 and 71 (±5) ka (Scholz
et al., 2007; Shanahan et al., 2008b). The duration of the event is less clear due to the excessive
erosion of the newly exposed crater walls and reduction in distance between the shore and 5B core
site (McKay, 2012), and this process likely caused unusually high sedimentation rates during this
interval. Thus, slight underestimation of sediment ages immediately following unit AI-1 may be
expected.

## 286 3.3. Sediment geochemistry

### 287 3.3.1. Mercury

Total Hg ($Hg_T$) in the bulk sediments of core BOS04-5B was measured using the RA-915 Portable
Mercury Analyzer with PYRO-915+ Pyrolyzer, Lumex (Bin et al., 2001) at the University of Oxford. For
this study, we analysed 165 samples spanning the composite depth interval 47.7 to 0 m, with an
average temporal resolution of ~0.6 ka between each sample (range: 0.01 to 5.85 kyr). Dry powdered
sample material (45–100 mg) was heated to ~700°C, volatilizing Hg in the sample. Atomic absorption
spectrometry of the gases produced during pyrolysis quantifies the total Hg content of the sample. Six
different quantities of standard material (paint-contaminated soil – NIST Standard Reference Material
® 2587) with a known Hg value of 290 ± 9 ng $g^{-1}$ were analysed to calibrate the instrument before
sample analysis, and then one standard for every 10 lacustrine samples. Long-term observations of
standard measurements *(n* = 390*)* for this instrument show average reproducibility (1 sigma) of 6% for
samples with ≥10 ng $g^{-1}$ Hg (Frieling et al. 2023). Four (2%) of the analysed samples contained very
low Hg contents (<10 ng $g^{-1}$), and likely have uncertainties ≥10 %. Details of standard runs are
included in an accompanying dataset.

### 302 3.3.2. Organic and inorganic carbon

Quantitative values for total organic carbon (TOC) and total inorganic carbon (TIC) content were
measured on the same powdered sample material also analysed for Hg, using a Strohlein Coulomat
(Jenkyns and Weedon, 2013) at the University of Oxford. Analytical reproducibility for this
instrument was ≤0.2 % based on repeat measurements, with a detection limit of ca. 0.1–0.2 %.
Powdered BOS04-5B sediment samples were split into two aliquots. Weights for aliquot 1 were
between 50–70 mg, and aliquot 2 between 90–120 mg. Prior to coulometric analysis, aliquot 2
samples were furnaced for 24 hours at 420°C in order to remove organic carbon fractions. Both
aliquots were then combusted in oxygen at 1220°C to break down the calcium carbonate and produce
carbon dioxide ($CO_2$), that was fed into a solution of barium perchlorate. By producing a change in
solution pH from an initial value of 10.0, back titration to the original pH using electrolysis provided a

measure of the amount of carbon originally present – quantified by the amount of electricity required to restore a pH of 10.0 and recorded in counts (Jenkyns, 1988; Jenkyns and Weedon, 2013). Counts obtained for aliquots 1 and 2 were used to calculate the total carbon (TC) content of each aliquot in wt.%, using the formula:

$$TC = \frac{total\ counts \times 0.2}{M} \quad \text{(eqn. 1)}$$

where M is the sample mass in mg.

TOC was calculated as follows:

$$TOC = TC_1 - TC_2 \quad \text{(eqn. 2)}$$

where $TC_1$ and $TC_2$ represent the TC values obtained for aliquots (1) and (2), respectively. $TC_2$ represents the TIC value for the sample. Our TOC curve was then compared with measurements previously obtained for BOS04-5B (on discrete samples): to assess the broader reproducibility of our results (**Fig. SF1**).

### 3.3.3. Authigenic carbonates

The BOS04-5B succession contains variable amounts of diagenetic carbonates, predominantly (mangano-)siderite (**Fig. SF2**) (McKay, 2012). Siderites commonly form in freshwater settings at shallow sediment depths under anaerobic (anoxic) conditions accompanied by organic-rich sediments (Armenteros, 2010; Sebag et al., 2018). However, they can also preclude accurate measurement of organic carbon content in lacustrine sediment via pyrolysis- or furnace-based methods, causing systematic overestimation of total organic carbon (TOC) due to the fact that thermal decomposition of siderite typically starts at temperatures <420°C (Sebag et al., 2018): lower than the temperature used to remove the organic fraction on the Coulomat (Jenkyns, 1988). To assess whether siderite-associated carbon had an appreciable impact on the TOC measurements, we also analysed the carbon release from sixteen BOS04-5B samples spanning a range of low to high XRF-derived Mn counts (i.e., covering the possible range of (mangano-)siderite contents) using a weak acid (warm 5% HCl) treatment, following established methodologies (**ST5**; Brodie et al., 2011; Vindušková et al., 2019). Comparison of acid-treated and furnaced samples showed no systematic offset nor a clear correlation with the Mn counts from XRF data (**Fig. SF2c**), suggesting that the carbon release from siderite did not appreciably bias TOC measurements.

### 3.3.4. Scanning X-Ray fluorescence

The Hg data for core BOS04-5B generated in this study are paired with unpublished x-ray fluorescence (XRF) data for Si, Ti, K, Mn, Ca, Fe, Rb, Sr, S, and Al (McKay, 2012). The bulk elemental composition of the core was quantified using the Itrax® scanning XRF analyser at the

Large Lake Observatory at the University of Minnesota. Core material covering the upper ~159 m
(~500-kyr) of BOS04-5B was analysed at 1-cm-resolution with 60 sec count times, and a Mo X-ray
source run at 30 kV and 20 mA. To mitigate any effects arising from changes in the physical
properties of the BOS04-5B sediments (e.g. compaction) and/or the measurement times, we applied
a centred-log ratio (clr) transformation to the measured XRF values. The centred log-ratio (clr) values
are calculated by dividing the intensities of an element by the average of the intensities obtained on
all selected elements, and are dimensionless such that positive values are generated for elements
with high intensities, and vice versa (Bertrand et al., 2024). Therefore, elements (X) subject to this
transformation are presented as their centred-log ratio value ($X_{clr}$).

## 3.4. Mercury normalization

It is common practice to assess both total Hg concentration ($Hg_T$) and normalised Hg (Hg/X) with the
aim to reduce, at least partially, the potential impact of variability in a dominant carrier/host phase (X)
on $Hg_T$ (Sanei et al., 2012; Shen et al., 2020). Organic matter (here expressed as TOC) is commonly
considered the primary host phase of sedimentary Hg (Ravichandran, 2004). However, variability in
$Hg_T$ may also be associated with variability in the abundance of detrital minerals, usually detected by
a correlation between Hg and detrital elements such as Al or K (Paine et al., 2024; Them et al., 2019),
and very rarely in sulphate-limited (lacustrine) sediments, sulphides (Benoit et al., 1999; Han et al.,
2008). Exploration of Hg signal variability relative to distinct shifts in the abundance, contribution
and/or sources of host phases can therefore elucidate the timing and magnitude of shifts in lake
hydrology, sedimentation regime, and geochemistry, and whether these are connected to changes in
the Hg cycle or sediment composition changes (Paine et al., 2024).
To isolate the effects of local depositional and/or transport processes on Hg signals recorded in the
sediments of Lake Bosumtwi, we normalised $Hg_T$ values to organic matter (TOC) and detrital mineral
abundance estimated from clr-transformed potassium intensities ($K_{clr}$); with the assumption that the
strongest positive-sloped linear correlation with $Hg_T$ among these elements signals the most likely
dominant impact of host phase variability in that section of the core. To account for differences in
resolution between Hg and XRF data, $K_{clr}$ values were averaged to obtain a $K_{clr}$ value corresponding
to the interval covered by each discrete Hg sample (~0.5 cm thickness).

## 3.5. Mercury accumulation

The total Hg mass accumulation rate ($Hg_{AR}$) in core BOS04-5B was calculated from:

$$Hg_{AR} = Hg_T (DBD \times SR) \qquad \textit{(eqn. 3)}$$

where $Hg_{AR}$ is in mg m$^{-2}$ kyr$^{-1}$, $Hg_T$ is the total mercury concentration (mg g$^{-1}$), DBD is the dry bulk
density (g m$^{-3}$), and SR is the sediment accumulation rate (m kyr$^{-1}$). Values for $Hg_{AR}$ are also
calculated with respect to the median age estimate for each sample. We do not present maximum
and minimum $Hg_{AR}$ values here, but note that uncertainties increase with depth due to increasing
uncertainties in sedimentation rates calculated based on the BOSMORE7 age model and average
~0.08 cm yr$^{-1}$ (0.02–0.3 cm yr$^{-1}$).
DBD values were calculated using the formula:
$$DBD = M_{solid}/V_{total} \qquad \text{(eqn. 4)}$$
where $M_{solid}$ is the mass of dry solid material (g) in each sample, and $V_{total}$ is the volume of each
respective sample (0.5 cm$^3$). To calculate $M_{solid}$, the proportion of clastic material was multiplied by an
assumed grain density value (2.6 g cm$^{-3}$) representative of a mixture of common sedimentary
minerals (e.g. quartz, clay minerals, clastic; typically range of 2.6 to 3 g cm$^{-3}$) and the total volume.
The proportion of clastic material was calculated by first accounting for the proportion of water and
organic matter (using loss-on-ignition; McKay (2012) and Shanahan et al. (2013) in each sample and
assuming the residual was all clastic material. DBD values generally increase with core depth,
reflecting the impact of increasing sediment compaction and dewatering with age (Shanahan et al.,
2013). Calculated values of DBD average 1.15 g cm$^{-3}$, and so are broadly consistent with
measurements taken from other African lake sediment successions of similar age (<100 ka),
composition (silty clays between 0.6–1.1 g cm$^{-3}$), and structure (high porosity) (e.g., Cohen et al.,
2016; Scholz et al., 2007).

## 3.6.     Statistical analyses

Two statistical analyses were used in order to more quantitatively explore the timing, and expression
of signals recorded in the BOS04-5B $Hg_T$ dataset. First was a simple linear Pearson's correlation
analysis (**ST6**), from which correlation coefficients (r) were calculated to indicate the direction and
strength of the association between Hg (n = 157), and a suite of geochemical proxies also measured
in the core. Second was a change point analysis (**ST7**), to determine whether distinct changes in
mean values of $Hg_T$ occur within the record using PAST v.4.16 software  (Hammer et al., 2001). This
software employs a Bayesian Markov chain Monte Carlo (MCMC) approach, which was run on default
settings with a total of 1 million MCMC simulations were run for each test and a maximum of ≤10
changepoints. The extent to which similar processes influenced the concentration of $Hg_T$, TOC, K, and
detrital matter were also explored, and correlations were subdivided based on visual examination of
the $Hg_T$ records (and supported by changepoint analyses).

# 4. Results

Core BOS04-5B from Lake Bosumtwi shows distinct fluctuations in total sedimentary Hg
concentration ($Hg_T$) throughout the ~47 m succession. Values range from 10 to 370 ng g$^{-1}$ (median:
58 ng g$^{-1}$) (**Fig. 2**). The $Hg_T$ curve broadly tracks that of TOC; the latter showing similarly pronounced
variability ranging between 0.1 to 23 wt. % (median: 6.5 wt. %) (**Fig. 2**), and peaking between 5.2 and
3 m depth. Calculated Hg accumulation rates ($Hg_{AR}$) do not follow the same pattern as $Hg_T$ and TOC.
Ranging between 2.9 and 460 mg m$^{-2}$ kyr$^{-1}$, calculated values instead broadly track the sedimentation
rate curve presented in **Figure 2**. Large peaks in $Hg_{AR}$ are visible between 8 and 6 m depth and then
again between 2 and 0 m, and these $Hg_{AR}$ peaks are both coeval with reductions in TOC below ~10
wt.%. The lowest $Hg_{AR}$ values are recorded in the lower core section between ~47 and 34 m depth.
Changing Hg signals in Lake Bosumtwi correspond to measurable changes in lake sedimentation.
From ~47 to 32 m depth, low amplitude, muted variability in both $Hg_T$ and $Hg_{AR}$ corresponds to a
homogeneous sequence of silty-clay sized material generally depleted in TOC, S, and high in detrital
materials. No clear changes in $Hg_T$ nor $Hg_{AR}$ are visible during AI-1 (34 – 32 m core depth), however,
variability in Hg concentration increases immediately following this interval. From ~32 m to the core
top, sediments show a progressive increase in $Hg_T$ punctuated by several clear peaks, and more
pronounced fluctuations in $Hg_{AR}$ (**Fig. 2**). This shift in Hg behaviour tracks a broad increase in the
organic content of the core, reflected by increasing TOC and decreasing clastic material percentages
(**Fig. 2**). The clearest expression of this correspondence is seen between 5.2 and 3 m, whereby the
highest $Hg_T$ values correspond to the organic-rich sapropel Unit S1.
Studying time-resolved changes in lake sediment Hg concentration provides a valuable opportunity to
study changes in the pre-industrial Hg cycle, how these changes translate to measurable sedimentary
signals, and their links to local and regional-scale environmental variability (Cooke et al., 2020). Two
mechanisms emerge as plausible drivers of Hg variability in Lake Bosumtwi (**Fig. 2**). First is organic
matter (host) availability, and second is external change in net Hg input to the system.

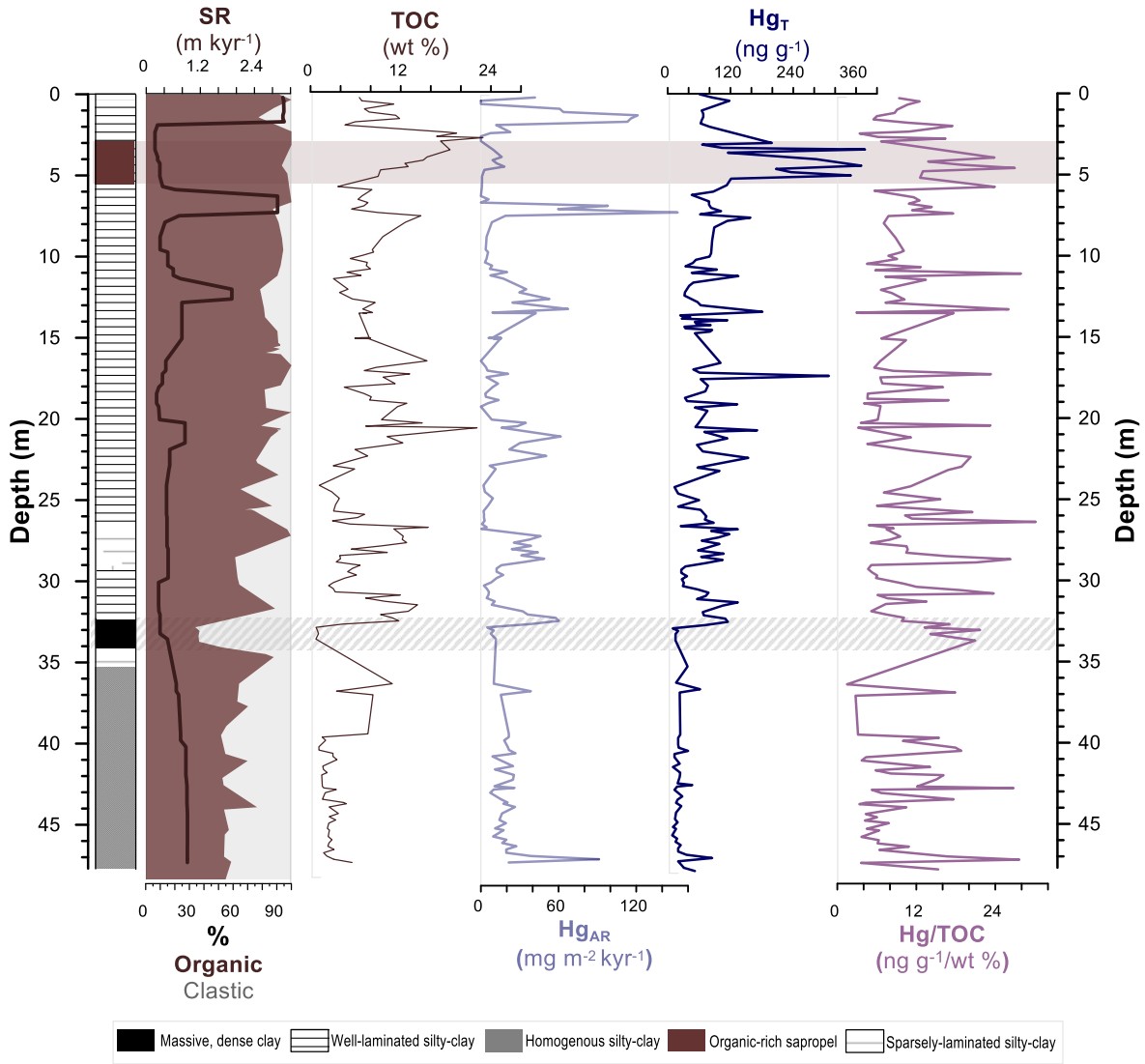

**Figure 2:** Depth-resolved profiles of total organic carbon (TOC), total Hg (Hg_T) and Hg accumulation rate (Hg_AR) profiles obtained for core BOS04-5B from Lake Bosumtwi in this study, relative to key lithofacies and sedimentological data including records of sedimentation rate (SR; *this study*), and the proportion of biogenic to terrigenous material (% organic) within the core (McKay, 2012). A distinct lake low stand referred to as Arid Interval 1 (AI-1) based on seismic profiles and sedimentological data is marked between 33.5 and 32.8 m depth (grey dashed shading; McKay, 2012; Scholz et al., 2007). Limited samples were available between ~39 and 34 m depth (**Fig. SF1**). Sapropel layer Unit S1 is marked between 3–5.5 m depth (brown shading; Russell et al., 2003; Shanahan et al., 2008a; Talbot and Johannessen, 1992). We also present ratios of Hg_T to TOC, following evidence for a positive correlation between the two compounds (r = 0.64; r$^2$ = 0.42) (see **section 4.1**).

## 4.1.    Lacustrine host phases of mercury


An overall positive association between Hg_T and TOC (r = 0.64; r$^2$ = 0.42) suggests that Hg variability
may be associated with organic carbon variability in Lake Bosumtwi. However, it is noteworthy that
detrital materials (e.g., K$_{clr}$) show negative correlations with both TOC (r = -0.59; r$^2$ = 0.34) and Hg (r =
-0.57; r$^2$ = 0.32) so that the Hg-TOC correlation may reflect, in part, a correlation imposed by variable
clay-dilution of both Hg and TOC. Moreover, these correlations are all significant at p <0.001 (unless
stated otherwise). The broad statistical link between Hg and TOC is supported by evidence for large
$Hg_T$ and $Hg_{AR}$ peaks in core sections containing high TOC concentrations, most markedly in the upper
sections (**Fig. 2**), and the relationship between $Hg_T$ and TOC also strengthens following deposition of
AI-1 (**Fig. 3b**). However, the highest $Hg_T$ values are not always recorded in the most TOC-enriched
sediments, nor are TOC-depleted sediments also depleted in $Hg_T$ (**Fig. 2**). Dilution of Hg by organic
matter is unlikely to be the cause (Machado et al., 2016), nor can shift from an organic to detrital-
dominated host-phase regime account for these signals, given that intervals characterised by an
overall negative Hg and TOC correlation are coeval with similarly negative values for $Hg_T$ and $K_{clr}$
(**Fig. 3, SF4**). More likely is that they reflect changes in net Hg flux to the system, and hence the
amount of Hg being supplied to (and sequestered in) Lake Bosumtwi.
A negative overall correlation between $Hg_T$ and $K_{clr}$ is apparent throughout the record (r = -0.57; $r^2$ =
0.32; **Fig. 3b, SF4**). Other robust proxies for the proportion of detrital and autochthonous components
in biogenic-rich sediments include Fe, Ti, Rb, and Al (**Fig. SF5**) (Grygar et al., 2019). Strong
correlations between $K_{clr}$ and these detrital elements confirm this is likely also the case in Lake
Bosumtwi (**Fig. 3**), with enrichment of detrital materials in this core reflecting enhanced erosion and
sediment transport to the BOS04-5B drill site (McKay, 2012; Shanahan et al., 2012). Moreover, the
significant negative correlations between TOC, $Hg_T$, and all elements associated with detrital
components ($K_{clr}$, $Ti_{clr}$, $Rb_{clr}$, and $Al_{clr}$) (**Fig. 3**) suggest that detrital matter did exert a control on $Hg_T$.
However, instead of increasing Hg, the negative correlation with detrital material suggests that 'Hg-
depleted' detrital materials diluted the concentration of both Hg and its suggested host (TOC).
Variations in both the detrital and Hg flux may also explain the somewhat counterintuitive decoupling
of $Hg_T$ and $Hg_{AR}$ in some intervals, for example, between ~10 and 6 m depth (**Fig. 2**). Dilution-driven
alteration of the sedimentary Hg record may be a common feature for depositional systems where
supply of Hg is ultimately limited by atmospheric inputs (e.g., Chede et al., 2022).
Strong correlations between $Hg_T$ and $Mn_{clr}$, and $Hg_T$ and $Fe_{clr}$ (redox sensitive elements) are also
absent in BOS04-5B (**Fig. 3**). The majority of the examined record is marked by the presence of
laminations, suggesting anoxic conditions dominate throughout (Shanahan et al. 2008). Although this
means that redox changes were likely subtle, the coeval presence of siderite has potential
implications for Hg through, for example, it's influence on Hg reduction (e.g., Ha et al. 2017). Fe is a
major component of siderite, but it is also an important component of the detrital material that washes
into Lake Bosumtwi and also potentially other redox-sensitive minerals precipitated in the lake and
sediment pore waters (Shanahan 2006; Shanahan et al., 2008; 2009). In this record, we therefore test
the relation between $Hg_T$ and Mn peaks. Pronounced Mn enrichments signal periods of
(mangano)siderite formation, and this constitutes the clearest indicator of (subtle) pore water redox
changes (McKay, 2012; Shanahan et al., 2008). Moreover, siderite specifically may be involved in Hg
cycling through its potential to reduce Hg (e.g., Ha et al. 2017). Overall, the lack of evidence for
substantial redox changes from ubiquitous laminations and the absence of a strong correlation
between Mn and Hg$_T$ suggests that Hg concentrations in Lake Bosumtwi were not appreciably
influenced by changes in redox conditions, nor the diagenetic effects signalled by these elements.

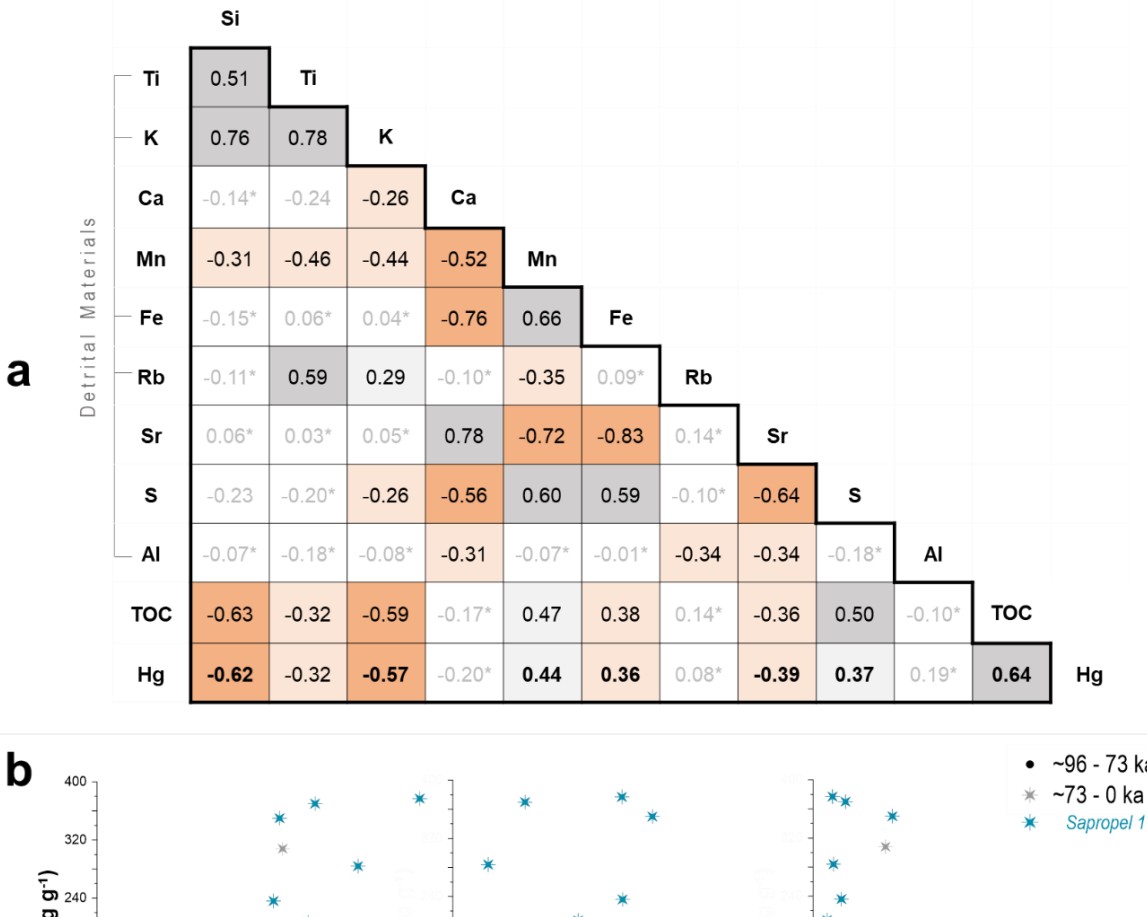

**Figure 3: (a)** Full-core correlation (Pearson's r) matrix for Hg, total organic carbon (TOC) (this study), and a suite of clr-transformed XRF data measured in BOS04-5B by XRF (McKay, 2012). Sample size (n) was 157 for each analysis, and >50% of the assessed trace element combinations were significant (p<0.01). Those combinations that were not are marked with an asterisk (*). Grey shading marks positive correlations (light: >0.25, dark: >0.5), and orange shading marks negative correlations (light: <-0.25, dark: <-0.5). Unshaded boxes mark weak/negligible correlations (between 0 and 0.25, and 0 and -0.25), with values greyed-out for clarity. All remaining values are presented with black text, with those in this range related to Hg in the boldest type. **(b)** Comparison of relationships in Lake Bosumtwi between ~96 and 73 ka (black circles), and between ~73 and 0 ka (stars). Relationships presented here are between Hg$_T$, total organic carbon (TOC), detrital minerals (estimated by potassium (K)) concentrations; McKay (2012), and principal component 1 (PC1) of the BOS04-5B XRF data, in which 39% of total variance is strongly associated with terrigenous elements (McKay, 2012). R (r) and r-squared (r$^2$) values for each interval are also given, and Student's t-testing showed that the

significance of all correlations were significant at p<0.01 Stars marked in teal correspond to deposition of sapropel unit 1 (S1) in BOS04-5B.

## 5. Discussion

### 5.1. Environmental drivers

Time resolved $Hg_T$ and $Hg_{AR}$ profiles generated from the sediments of Lake Bosumtwi show two broad periods of differing Hg behaviour: (1) ~96 – 73 ka (low $Hg_T$ and $Hg_{AR}$) and (2) ~73 – 0 ka (moderate/high $Hg_T$, and large fluctuations in $Hg_T$ and $Hg_{AR}$) (**Fig. 4**). Each corresponds to different lake level evolution trends with broadly decreasing lake level between ~96 and 73 ka (although with a substantial rise between 95 and 80 ka), and rising from ~73 to 0 ka (**Fig. 4**). Lake Bosumtwi's hydrology is controlled by a balance between direct precipitation and runoff with water removal limited almost entirely to evaporation; exceptions being rare transient overspilling events (Turner et al., 1996). Taking this unique hydrology into account, our discussion below explores how different environmental processes relate to changes in Hg behaviour during these two distinct intervals, and how the significance of these processes may have changed through time.

#### 5.1.1. Arid conditions (~96 to 73 ka)

Both $Hg_T$ and $Hg_{AR}$ show muted variability between ~96 and 73 ka (**Fig. 4**). The presence of more clastic-rich/organic-depleted sediment (**Fig. 4**), and reductions in tree pollen are both typical of a savannah-dominant, more open landscape (**Fig. 4)**, and so suggest generally arid conditions within the lake and its catchment prior to ~73 ka. These conditions would favour pronounced reductions in lake level (McKay, 2012; Miller et al., 2016; Scholz et al., 2007), and are consistent with a 24 – 38% reduction in local rainfall as estimated by water balance modelling) (Shanahan et al., 2008b). Reductions in lake level could facilitate an increase in water column vertical mixing, ventilation of bottom waters, more efficient breakdown of organic matter, and simultaneous sediment dilution by a sudden increase in eroded material fluxes (McKay, 2012; Scholz et al., 2007; Shanahan et al., 2012) – all could lead to a reduction in organic matter (host-phase) concentration. Indeed, several meromictic lakes have shown reduced organic matter content as a function of better ventilation and lower productivity during 'shallow' conditions (Katsev et al., 2010; Schultze et al., 2017), and new evidence suggests that changes in organic matter oxidation may produce comparably distinct changes in Hg sequestration (Tisserand et al., 2022).

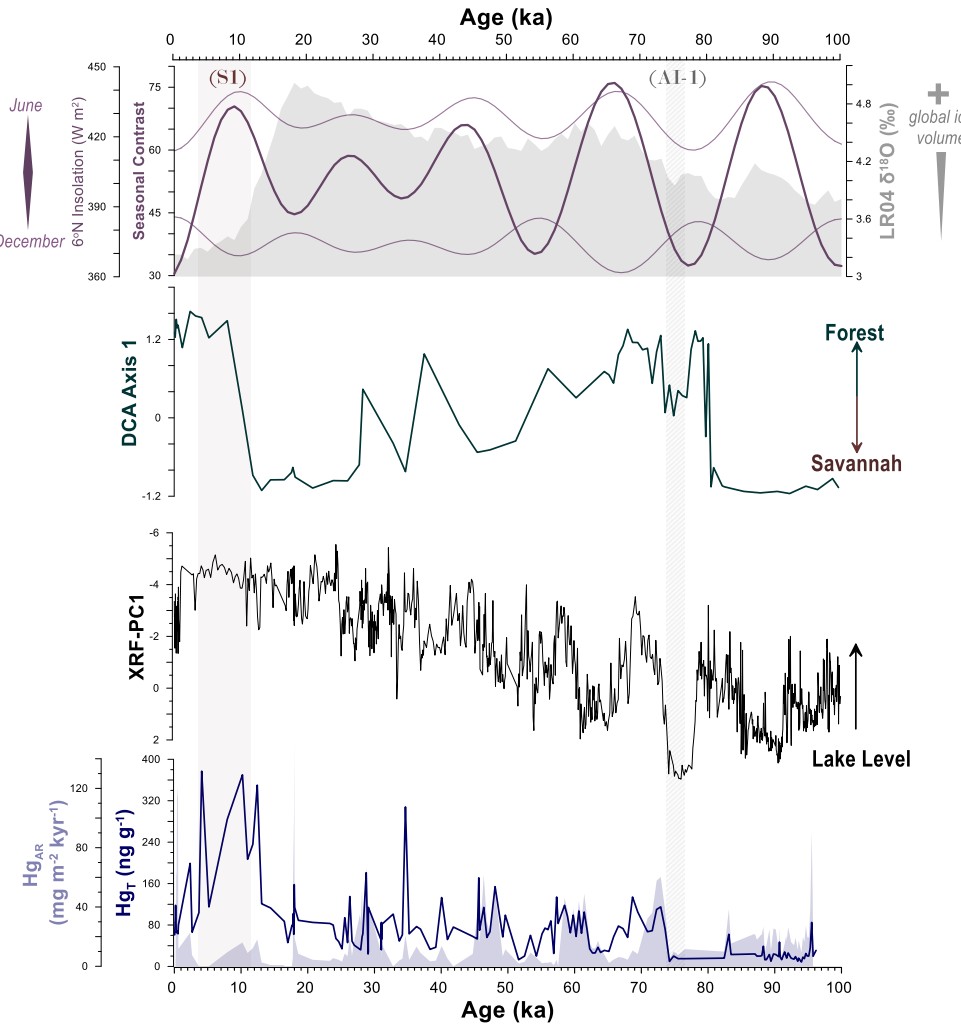

**Figure 4:** Comparison of key proxy datasets. Included are (from bottom to top), total mercury (Hg$_T$) and mercury accumulation rate (Hg$_{AR}$) for Lake Bosumtwi from this study (see **section 5.1**), the first principal component (PC1) of the BOS04-5B XRF data (39% of total variance, interpreted as an indicator of lake level changes; McKay, 2012), forest (woody) taxa abundance (presented as DCA Axis 1; Gosling et al., 2022a; Miller et al., 2016), and insolation at 6°N (location of Lake Bosumtwi) in June (summer) and December (winter) calculated following the astronomical solution presented by Laskar et al. (2004) (accessed via. https://vo.imcce.fr/insola/earth/online/earth/online/index.php), and used to calculate seasonal insolation contrast. Also shown is a record of benthic foraminiferal calcite δ$^{18}$O (‰) derived from the LR04 global stack) as a proxy for ice volume, with cold glacial stages defined by high δ$^{18}$O ratios (Lisiecki and Raymo, 2005a). Proxy data are all presented on the BOSMORE7 chronology. Unit AI-1 is marked between 33.5 and 32.8 m depth (grey shading; Brooks et al., 2005; Scholz et al., 2007), and sapropel layer Unit S1 is marked between 3–5.5 m depth (brown shading; Shanahan et al., 2012, 2006).

The absence of a clear change in Hg$_{AR}$ between ~96 and 73 ka might also reflect changes in the
balances of Hg cycling in the lake. Lake Bosumtwi is a hydrologically closed system that receives
>80% of its water from rainfall directly on the surface, meaning its hydrology and sedimentation
regime is extremely sensitive to variability in precipitation and precipitation-evaporation balance
(Shanahan et al., 2007; Turner et al., 1996). Thus, low Hg$_T$ and Hg$_{AR}$ values may reflect a reduction in
wet deposition of atmospheric Hg at the Lake Bosumtwi site by precipitation, while Hg evasion back to

the atmosphere remains high due to evaporation in the consistently warm, tropical temperatures (Schneider et al., 2023). Depletion of sedimentary Hg during drier climate intervals are documented in several other late Quaternary-age records, where they are interpreted as signs of a net reduction in Hg input relative to loss/evasion (e.g., Hermanns and Biester, 2013; Pompeani et al., 2018; Schneider et al., 2020; Schütze et al., 2021, 2018).

Desiccation of Lake Bosumtwi between ~75 and 73 ka (AI-1) corresponds to evidence for severe depletion of organic matter, and enrichment of detrital materials within the sediments (**Fig. 2**). Although a detailed characterization of local soil and bedrock Hg contents is currently lacking for Lake Bosumtwi, these changes in sediment composition (lower TOC, higher $K_{clr}$) and low overall sedimentation rates are unaccompanied by coeval changes in $Hg_{AR}$ (**Fig. 2**). In certain cases, one would typically expect that the near-complete desiccation of a steep-sided lake would 'focus' trace metals (including Hg) at the central coring site, particularly during lake recessions following erosion of exposures around the crater rim (Blais and Kalff, 1995; Engstrom and Rose, 2013). However, evidence for low Hg burial both prior to and during AI-1 in Lake Bosumtwi suggests that over multiple millennia, changes in Hg supply to the BOS04-5B drill site were predominantly driven by atmospheric inputs, with minimal contribution from catchment-sourced materials.

### 5.1.2. Humid conditions (~73 to 0 ka)

The magnitude and frequency of variability in $Hg_T$ visibly increases at ~73 (±5) ka (**Fig. 4**). The quantitative significance of this shift is supported by changepoint analysis of the BOS04-5B dataset, which demonstrates a clear and step-wise shift in mean $Hg_T$ values between ~75 and 73 ka (**Fig. SF3**). It also occurs in conjunction with an increase in the lake's water level (**Fig. 4b**), which is corroborated by a statistically significant relationship between PC1 (lake level indicator; McKay, 2012), and $Hg_T$ in our record (**Fig. 3b** - r = -0.36; $r^2$ = 0.13). Furthermore, changing sedimentary TOC, terrigenous material, and pollen concentrations all corroborate a broad increase in local moisture availability, temperature, and humidity following deposition of the AI-1 unit (**Fig. 4**): a signal that coincides with the transition into the warmer Holocene interglacial, marked by reduced global ice volume and increased sea surface temperatures in the North Atlantic (McKay, 2012; Scholz et al., 2007; Shanahan et al., 2008b). Our data also shows a simultaneous increase in $Hg_T$, and a decrease in detrital material concentrations following ~73 ka (**Fig. 4**), suggesting that Hg supply temporarily exceeded the diluting effects of clastic materials following lake level rise.

Lake deepening generally increases water column stratification, limiting the effects of vertical transport processes such as turbulent energy generated by surface winds and currents (Gulati et al., 2017). Deeper, more anoxic conditions are also typically associated with more effective organic carbon burial (Gulati et al., 2017; Schultze et al., 2017), coupled with more distinct formation of distinct laminations (Zolitschka et al., 2015) and precipitation of authigenic carbonates such as siderites (Swart, 2015). Given that elevated $Hg_T$ and $Hg_{AR}$ correlate most closely with TOC enrichment in Lake Bosumtwi following ~73 ka (**Fig. 3b**), this could suggest that Hg drawdown was

moderated by an increase in organic matter availability and preservation, as an indirect function of
bottom water deoxygenation. Evidence for an inverse relationship between sedimentary Hg
concentration and hypolimnion oxygen content has been identified in a number of meromictic lake
systems across the world (e.g., Schultze et al., 2017; Tisserand et al., 2022), and provides further
support for our interpretation.
Geochemical and sedimentological evidence suggests near-permanent oxygen depletion in Lake
Bosumtwi following ~73 ka, which likely contributed to formation of a very organic-rich sapropel
formation between 12.4 and 3.7 ka. This unit contains clear $Hg_T$ enrichments relative to the rest of the
core (**Figs. 2, 4a, SF3**), is extremely rich in organic matter (~15-20%), and contains a high
concentration of blue-green algae *Anabaena* deposits (Russell et al., 2003). Sapropelic layers have
emerged as key sites of Hg enrichment from a suite of marine and lacustrine-based studies,
suggesting this may be due to changes in productivity, sediment oxygenation, and diagenetic
processes (e.g., Frieling et al., 2023; Gehrke et al., 2009; Jeon et al., 2020). Scavenging of Hg from
the water column by algae is also recognised as an important driver of Hg export to lacustrine
sediments; particularly in systems where primary productivity, organic matter production, and burial
capacity is high (Biester et al., 2018; Schütze et al., 2021; Outridge et al. 2007). These conditions are
met in Lake Bosumtwi following ~73 ka, meaning the observed changes in sedimentary $Hg_T$ could be
linked to elevated rates of scavenging, as a function of enhanced primary productivity.
In closed-basin lakes where fluxes of organic material from the catchment (e.g., soils and vegetation)
are minimal, measurable changes in sedimentary Hg concentration would require a simultaneous
increase in Hg fluxes to the system in order to counterbalance Hg depletion by scavenging,
methylation, or evasion back to the atmosphere (e.g., Bravo et al., 2017; Hermanns et al., 2013;
Outridge et al., 2007; Schütze et al., 2021). For Lake Bosumtwi these direct inputs may have come
from precipitation, and/or from increased flux of charcoal and associated release of Hg from
vegetation during wildfires into the lake following local wildfire events; the latter documented by
variations in the micro charcoal concentration of BOS04-5B (Gosling et al., 2021; Miller et al., 2016;
Miller and Gosling, 2014).
Data from Lake Bosumtwi suggests an increase in local precipitation following ~73 ka. Model and
proxy-based data show that the precipitation-evaporation balance is directly coupled to lake level in
this system, such that lake deepening occurs as a function of more rainfall (Shanahan et al., 2008b).
Proxy data generated from the BOS04-5B core suggest that progressively wetter conditions affected
the catchment following ~73 ka (e.g., Gosling et al., 2022a; Shanahan et al., 2008b). Paleoclimate
records based on the sediments of lakes Malawi (Tanzania), Bambili (Cameroon), Tanganyika
(Tanzania/Democratic Republic of the Congo), Chew Bahir (Ethiopia) and Chala (Tanzania) (e.g.,
Cohen et al., 2007; Foerster et al., 2022; Lézine et al., 2019; Scholz et al., 2007), and marine
sediment core material from the West African margin (**Figs. 1, 5**) (e.g., Kinsley et al., 2022;
Skonieczny et al., 2019) also pertain to a distinct regional-scale shift in hydro climate across tropical
sub-Saharan Africa at this point in time (**Fig. 5**). Specifically, a shift characterized by a distinct
moisture gradient favouring wetter conditions in the west of the continent relative to the east, which
was further amplified during the last glacial termination (e.g., Baxter et al., 2023; Gosling et al., 2022a;
Lupien et al., 2023). Therefore, the coeval increase in the frequency and amplitude of Hg enrichments
in Lake Bosumtwi, and associated rises in lake level, could indirectly reflect pronounced shifts in
hydroclimate across tropical sub-Saharan Africa.

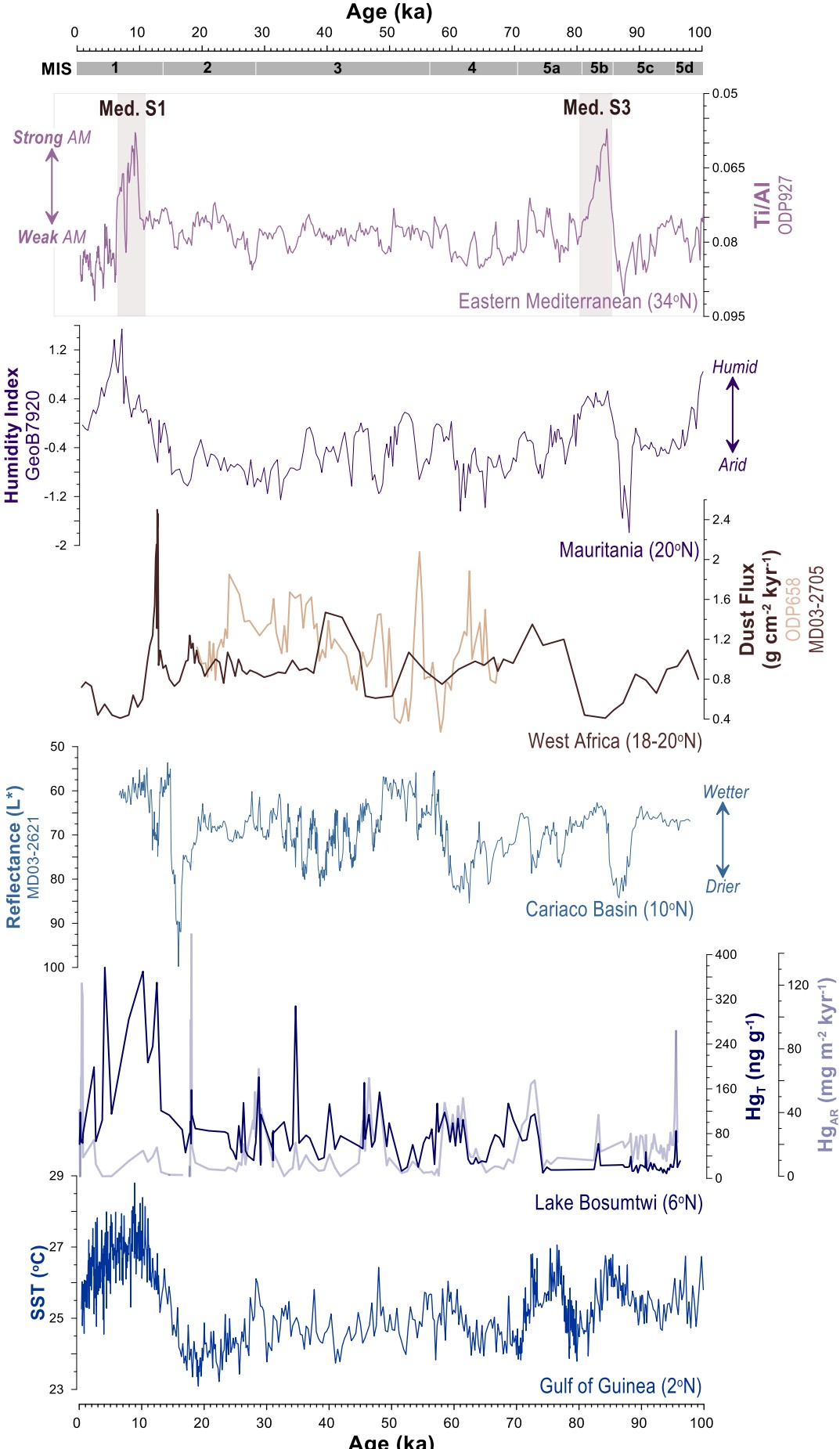

**Figure 5**: Records of total mercury ($Hg_T$) and mercury accumulation rate ($Hg_{AR}$) for Lake Bosumtwi generated by this study, compared with key paleoclimate records presented in order of latitude (physical locations shown in **Figure 1**), and known to be influenced by the WAM: sea-surface temperature (SST) reconstructed in core MD03-2707 from the Gulf of Guinea (Weldeab et al., 2007), sediment total reflectance (L*) in marine core MD03-2621 from the Cariaco Basin, Venezuela, as a proxy for hydrological conditions anticipated in light of ITCZ oscillations over West Africa (Deplazes et al., 2013), dust fluxes recorded in cores ODP658 (Cap Blanc; Kinsley et al., 2022) and MD03-2705 (; Skonieczny et al., 2019), and a continental humidity index of core GeoB7920-2 (Mauritanian seamount; Tjallingii et al., 2008) – all from offshore Mauritania. Finally, Ti/Al recorded in core ODP 927 from the Eastern Mediterranean as a record of riverine (low Ti/Al) versus aeolian (high Ti/Al) North African inputs to the Mediterranean basin, and thus African monsoon intensity (Grant et al., 2022, 2017). Mediterranean sapropels one (Med. S1) and three (Med. S3) are marked by light brown bars (Grant et al., 2016). Light grey bars mark marine isotope stages (MIS) defined by the LR04 benthic marine isotope stack (Lisiecki and Raymo, 2005b).

Volcanic and wildfire activity are both linked to the Hg cycle. Although Lake Bosumtwi is located in
close proximity to several highly productive volcanic regions, the resolution of BOS04-5B precludes
our ability to examine Hg emissions with respect to single eruption events (**Fig. SF7**), and eruption
record incompleteness coupled with time-transgressive changes in the global atmospheric Hg burden
both complicate the ability to unambiguously correlate enhanced volcanic emissions to greater Hg
deposition (**Fig. SF7**). Hydroclimate was also a key driver of changes in fire activity in tropical sub-
Saharan Africa during the late Pleistocene (Moore et al. 2022). However, although wetter climatic
conditions may be broadly associated with heightened fire activity due to associated increases in
terrestrial biomass, recent work has shown that discrete changes in precipitation can elicit notably
different fire responses between sites (e.g., Gosling et al., 2021; Karp et al., 2023). The influence of
biomass burning on the Hg record presented here appears similarly complex; despite being a well-
constrained factor in the Bosumtwi catchment, and evidence that wildfires are also a significant
source of Hg, accounting for ~13% of natural Hg (re-)emissions to the modern atmosphere (Francisco
López et al., 2022). Given that no clear relation is visible between $Hg_T$, $Hg_{AR}$, and two discrete macro-
(Kiely, 2023) and micro- (Miller et al., 2016) charcoal profiles generated from the BOS04-5B core, we
suggest that the effects of Hg emitted during wildfires also did not leave a clear imprint on Hg
variability in this record (**Fig. SF8**).

## 6. Synthesis and conclusions

This study combines new sedimentary Hg data from Lake Bosumtwi, Ghana, with proxy data from
archives across the African continent to explore whether hydroclimate has exerted a measurable
effect on regional Hg cycling over the past ~96-kyr. The resolution of the BOS04-5B record (~0.6 kyr
per sample) precludes a detailed assessment of more recent (<0.2-kyr), anthropogenic-driven
changes in local Hg cycling. However, this record is well suited for a broader exploration of patterns
and drivers of variability in sedimentary Hg concentrations in Lake Bosumtwi during the late

Pleistocene. Combining our results with existing data reveals two possible drivers of variability in $Hg_T$ and $Hg_{AR}$ in Lake Bosumtwi on these timescales: organic matter (host) availability, and local-scale changes in Hg input to the lake by precipitation (**Fig. 4**). Both are intrinsically coupled to the local hydroclimate by their link to the lake level, with higher lake levels typically corresponding to wetter conditions in the catchment, and deposition of more organic-rich sediments. **Figure 6** illustrates how selected environmental processes, under different environmental conditions, may have interacted with these two drivers to control Hg burial in Lake Bosumtwi between ~96 and 0 ka. Considered together, the evidence summarised in panels **(1), (2),** and **(2a)** all suggest that rates of Hg drawdown in Lake Bosumtwi, and indeed the signals retained in the sediment record, reflect changes in net Hg supply from the atmosphere.

Between ~96 and 73 ka (**Fig. 6, panel (1)**), generally arid conditions shifted the lake into a negative water balance. Not only could this have reduced the net flux of Hg to the lake by wet deposition (precipitation), but a negative water balance would also limit internal primary productivity and preservation, and so render less organic material available to sequester any Hg present in the system. Secondary dilution of Hg by detrital materials could have also lowered sedimentary Hg concentrations, with elevated delivery of terrigenous matter to the BOS04 site driven by exposure of the steep-sided crater walls during lake level lowering, and heightened soil instability due to widespread recession of catchment vegetation. All would persist (if not strengthen) during AI-1, and so could explain the lack of any measurable changes in $Hg_T$ and $Hg_{AR}$ during this time.

Following an extended period of aridity, net supply of Hg to the basin would be increased by precipitation following ~73 ka. This would simultaneously cause the lake to become deeper and more stratified (**Fig. 6, panel (2)**). As the bottom waters became more oxygen-depleted, more effective organic matter burial would simultaneously enhance Hg drawdown compared to detrital mineral supply; with higher lake levels, vegetation growth, and soil stabilization preventing exposure and erosion of the crater walls and soils surrounding the lake. Hence, this abrupt shift to humid (net-positive precipitation-evaporation balance) conditions in the Bosumtwi catchment could plausibly have driven an increase in sedimentary Hg concentrations and accumulation, by eliciting a pronounced rise in lake level as well as increasing the atmospheric Hg flux.

The processes described in panel (2) would be amplified further between ~15 and 4 ka (**Fig. 6, panel 2a**). Corresponding to Bosumtwi sapropel unit 1, this unit marks a distinct humid period characterised by anomalously high rainfall, and documented by proxy records across tropical sub-Saharan Africa (Shanahan et al., 2015). For a closed lake system such as Lake Bosumtwi, these wetter conditions would drive a sharp increase in lake depth, stratification, and scavenging in the water column – all could favour heightened Hg drawdown to the sediment. 'Flattening' of the Hg-TOC relationship during this interval also suggests that the Hg supply was (far) exceeded by the organic matter availability (**Fig. 3a**), and so elevated Hg supply by precipitation could explain why $Hg_T$ and $Hg_{AR}$ values are so unusually high (**Fig. 4a, 5**).

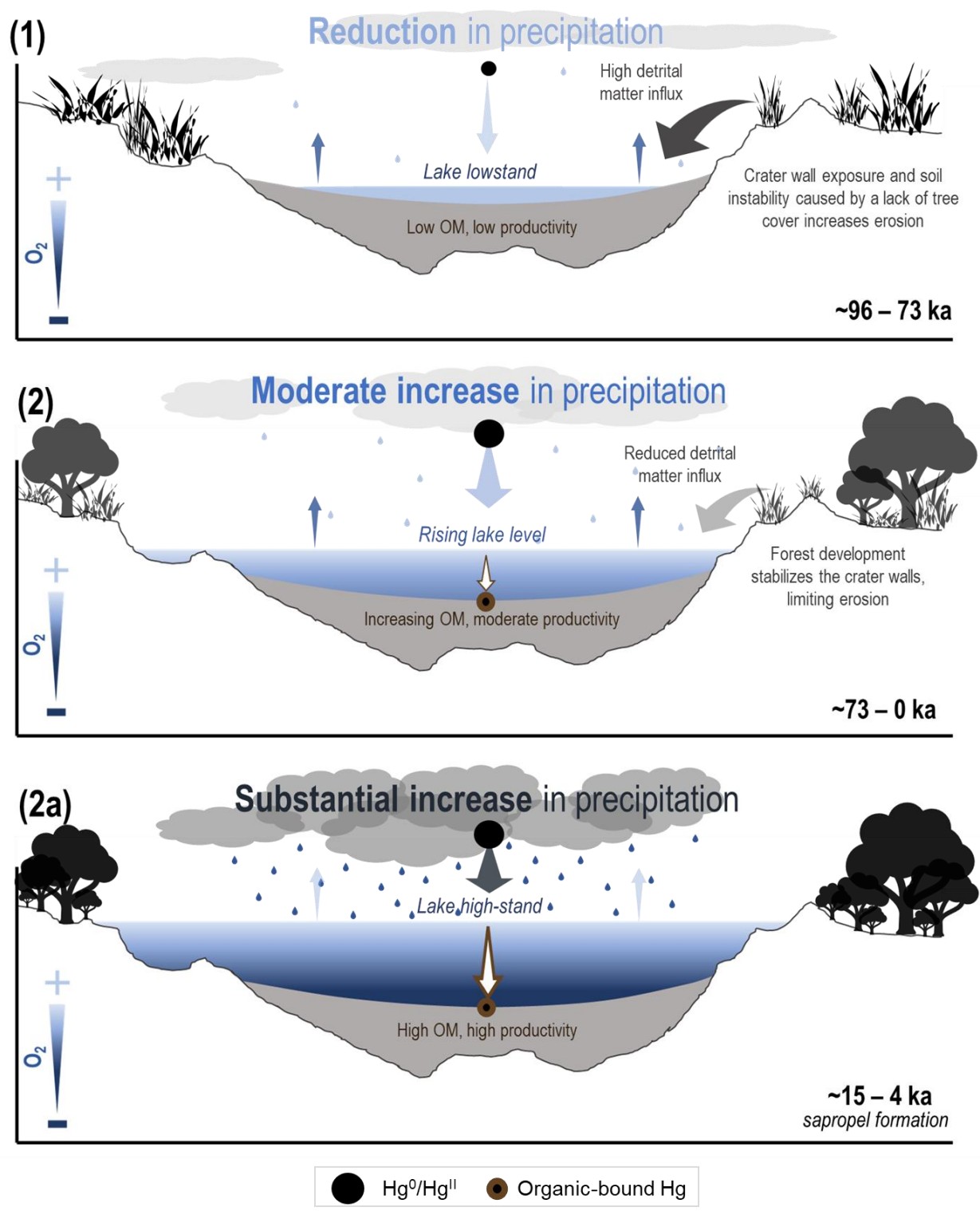

**Figure 6:** Schematic model depicting the processes that may control Hg flux, accumulation, and burial in Lake Bosumtwi under **(1)** arid (~93 – 73 ka), and **(2)** humid (~73 – 0 ka), environmental conditions. Panel **(2a)** depicts the very humid conditions that would be conducive to sapropel formation, such as those known to have occurred during the African Humid Period (~15 – 4 ka). Taken together, Hg fluxes increase during wet periods due to higher wet deposition directly to the lake relative to evasion, and/or by enhanced mobilization and transport of Hg from the

catchment. Hg sequestration can also be enhanced by OM-scavenging in the water column, and increased lake stratification (anoxia at lake floor). The opposite occurs during dry intervals.

Future research should seek better constraints on how basin-specific variations in sediment
composition, lake structure, and water balance may influence how sedimentary Hg signals are
preserved and interpreted. This is because all could produce diverse, and perhaps contrasting, results
between lake systems. For example, results from Lake Bosumtwi suggest that lakes with smaller
watersheds, simple morphology, and minimal hydrological connectivity to the catchment could be
suitable targets to study catchment and intra-lake depositional processes over multiple millennia.
However, there are currently too few records covering these timescales to say this with certainty, and
not all closed lakes record measurable changes in Hg composition corresponding to changes in local
hydroclimate (Lent and Alexander, 1996; Pompeani et al., 2018). Organic matter/host phase
availability also appears to represent just one of several possible processes governing Hg burial in
lacustrine systems, given these systems are more readily affected by short-term changes in erosion,
nutrients, water balance, and catchment hydrology (Paine et al., 2024; Schütze et al., 2021).
Lake Bosumtwi is a small, morphologically simple lake. However, the complexity shown by its
sedimentary Hg record suggests that identical stratigraphic signals are unlikely to be recorded in
separate lakes, even if they are dominated by one common process, mechanism, and/or structure.
Exploring the importance of hydroclimate for Hg cycling relative to different catchment to lake area
ratios, hydrology (e.g., endorheic (closed) versus exoreic (open)), and/or catchment structures (e.g.,
forest versus savannah) would undoubtedly help to better resolve processes acting on single
lacustrine and terrestrial successions, but also identify the systems that may more sensitively record
major changes in Hg cycling. Provided the hydrological component of the Hg cycle can be isolated,
better characterization of the processes impacting lacustrine Hg cycling could also allow this element
to be used as a proxy for hydroclimatic change in terrestrial archives. For example, measurement of
sedimentary Hg isotopes in low-latitude and/or closed lakes could help quantify the contribution of Hg
to the sediment from precipitation or dry deposition, shedding new light on key biogeochemical
reaction pathways, processes (e.g., mass-independent fractionation (MIF)), and responses to
changing local hydrology across a range of timescales (e.g., Blum et al., 2014; Gao et al., 2023; Yin
et al., 2024).
This study provides new and valuable evidence for long-term interactions between terrestrial Hg
cycling and hydroclimate, and demonstrates that hydroclimate may be a key driver of Hg cycling in
tropical lakes over millennial-timescales. The sparse number of continuous, pre-industrial Hg records
currently available for sub-Saharan Africa have historically limited the ability to understand if, or how,
hydroclimate may drive long-term (>$10^2$-year) variability in the Hg cycle (Schneider et al., 2023), and
subsequently how this relationship is represented in local and global ecosystem models (Cooke et al.,
2020; Obrist et al., 2018). Although this knowledge gap cannot be satisfied by a single record, study
of Lake Bosumtwi reinforces the value of these records for better characterization of the Hg behaviour
likely to be associated with projected future, monsoon-driven, hydroclimate variability (Chang et al.,
2022). In time, this could translate to better understanding of how the tropical Hg cycle may respond
to future, global-scale changes (Gustin et al., 2020; Schneider et al., 2023).

## Competing Interests

The contact author has declared that none of the authors has any competing interests.

## Acknowledgements

ARP, IMF, JF, and TAM acknowledge funding from European Research Council Consolidator Grant
V-ECHO (ERC-2018-COG-8187 17-V-ECHO). ARP thanks Christopher Scholz for provision of
sediment data, alongside James Bryson, Alex Dickson, Erdem Idiz, Francesco Pausata, Matt Jones,
and Victoria Smith for insightful discussions at various stages of manuscript preparation. Thanks also
go to Stephen Wyatt (University of Oxford) for analytical assistance throughout the study. All authors
thank members of the International Continental Scientific Drilling Program Lake Bosumtwi Drilling
Project: for their efforts in extracting and producing the sediment succession, and making the data
available for scientific use.

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
