# Peer review of "Evidence for millennial-scale interactions between Hg cycling and hydroclimate from Lake Bosumtwi, Ghana"

_EGUsphere, 2024_

## Author Response (AR1)

Please find below our detailed responses (in blue) to comments given by Reviewers #1 and #2, where the original reviewer comments are repeated here in black for clarity and completeness.
* * *
**REVIEWER #1**

Paine and coauthors present a new Hg record in a unique sedimentological archive from Lake Bosumtwi that records West African hydroclimate over the last 96 thousand years. The study is partly honing and improving upon the Hg proxy for paleoclimate reconstructions and partly discussing hydroclimate history of the region at various timescales. The study is well executed and very well written, but the impact of the new record is a bit limited by the uncertainty in Hg driving mechanisms, but also the pre-existing reconstructions of lake level that may be more powerful for the paleoclimate framing. Please find comments, technical edits, and suggestions for improving the figures below.

> We sincerely thank reviewer #1 for taking the time to provide feedback on our manuscript, and for their kind words regarding its presentation. In the response below and in our revised manuscript, we outline how we will ensure their comments and suggestions are thoroughly considered and addressed where necessary.

**Moderate comments:**

The split at 73 ka should be tested with statistical tools such as change point analysis to see how significant it is, rather than a visual analysis. Much of the discussion hinges on the differences before and after this time, so it should be bolstered by significance testing.

> This is a great suggestion, and so we have proceeded to carry out the change point analyses on our record. To test the significance of the shifts in $Hg_T$ we identified in the manuscript, we used Paleontological Statistics software (PAST) v.4.16 to apply a change point analysis function to the BOS04-5B $Hg_T$ data (Hammer et al., 2001). The results of this analysis are shown below, and will be incorporated into the accompanying supplementary information file.

[Figure]

**Figure S3**: The average changepoint model for BOS04-5B displayed as a purple curve, superimposed onto the original $Hg_T$ values. The abruptness of the curve indicates the extent to which the MCMC simulations (n = 1000000) agree on the changepoint positions, where greater smoothing indicates greater variance between simulations. Unit AI-1 is marked between 33.5 and 32.8 m depth (grey shading; Brooks et al., 2005; Scholz et al., 2007), and sapropel layer Unit S1 is marked between 3–5.5 m depth (brown shading; Shanahan et al., 2012, 2006).

> We will also add the following information to our revised submission:

> (1) A description of statistical methods employed by our study in **section 3**:

**Line 391**: "*3.6.    Statistical analyses*
*Two statistical analyses were used in order to more quantitatively explore the timing, and expression of signals recorded in the BOS04-5B Hg$_T$ dataset. First was a simple linear Pearson's correlation analysis, from which correlation coefficients (r) were calculated to indicate the direction and strength of the association between Hg (n = 157), and a suite of geochemical proxies also measured in the core. Second was a change point analysis, to determine whether distinct changes in mean values of Hg$_T$ occur within the record using PAST v.4.16 software (Hammer et al. 2001). This software employs a Bayesian Markov chain Monte Carlo (MCMC) approach, which was run on default settings with a total of 1 million MCMC simulations were run for each test and a maximum of ≤10 changepoints. The extent to which similar processes influenced the concentration of Hg$_T$, TOC, K, and detrital matter were also explored, and correlations were subdivided based on visual examination of the Hg$_T$ records (and supported by changepoint analyses). The significance of all correlations was assessed using a Student's t-test, which showed that ~75% of the assessed geochemical combinations were significant at p<0.01.*"

(2)  Explicit reference to the changepoint analyses in **section 4**:

**Line 516**: "*The magnitude and frequency of variability in Hg$_T$ visibly increases at ~73 (±5) ka (**Fig. 4**). The quantitative significance of this shift is supported by changepoint analysis of the BOS04-5B dataset, which demonstrates a clear and step-wise shift in mean Hg$_T$ values between ~75 and 73 ka (**Fig. S3**). It also occurs in conjunction with an increase in the lake's water level (**Fig. 4b**),*"

**Line 542**: "*This unit contains clear Hg$_T$ enrichments relative to the rest of the core (**Fig. 2, 4a, S3**)…*"

3) A clear statistical test of the relationship between Hg$_T$ and the first principal component (PC1) of the BOS04-5B XRF data (proxy for lake level; McKay, 2012) both prior to, and following ~73 ka (*see below*).

4) Explicit listing of r-values in **Figure 3b:** to further clarify the change in significance and sign of correlation of Hg, TOC, detrital matter, and lake level relationships following ~73 ka.

[Figure]

**Figure 3: (b)** Comparison of relationships in Lake Bosumtwi between ~96 and 73 ka (black circles), and between ~73 and 0 ka (stars). We first assess the Hg$_T$ record for this lake relative to two potential host-phases: total organic carbon (TOC) values measured in this study, and detrital minerals (estimated by potassium (K)) concentrations measured by McKay (2012). We also test the relationship between Hg$_T$ and first principal component (PC1) of the BOS04-5B XRF data, in which 39% of total variance is associated with terrigenous elements. PC1 was consequently interpreted as an indicator of lake level changes (McKay, 2012). R (r) and r-squared (r$^2$) values for each interval are also given. The significance of all correlations were assessed using a Student's t-test, which showed that all three combinations were significant at p<0.01. Stars marked in teal correspond to deposition of sapropel unit 1 (S1) in BOS04-5B.

The pattern in Hg and pollen look a lot like ice volume…

A more explicit comparison of our record with a proxy for global ice volume would certainly add valuable context to our discussion. To do this, we will add a record of benthic foraminiferal calcite δ$^{18}$O (‰) derived from the LR04 global stack to **Figure 4**, where cold glacial stages characterized by high ice volume are defined by high δ$^{18}$O ratios.

[Figure]

**Figure 4:** Comparison of key proxy datasets. Included are (from bottom to top), total mercury (Hg$_T$) and mercury accumulation rate (Hg$_{AR}$) for Lake Bosumtwi from this study, chosen as the most appropriate proxies for Hg variability in this core (see **section 5.1**). The first principal component (PC1) of the BOS04-5B XRF data (39% of total variance) is strongly associated with terrigenous elements, and so interpreted as an indicator of lake level changes (McKay, 2012). Forest (woody) taxa abundance (presented as DCA Axis 1; Gosling et al., 2022a; Miller et al., 2016). Lack of data for woody taxa presence is assumed to imply a savannah-dominated regional landscape. Insolation at 6°N (location of Lake Bosumtwi) in June (summer) and December (winter) are calculated following the astronomical solution presented by Laskar et al. (2004) (accessed via. https://vo.imcce.fr/insola/earth/online/earth/online/index.php), and used to calculate seasonal insolation contrast at the Bosumtwi site since ~100 ka. Also shown is a record of benthic foraminiferal calcite δ$^{18}$O (‰) derived from the LR04 global stack) as a proxy for ice volume, with cold glacial stages defined by high δ$^{18}$O ratios (Lisiecki and Raymo, 2005). Proxy data are all presented on the BOSMORE7 chronology. Unit AI-1 is marked between 33.5 and 32.8 m depth (grey shading; Brooks et al., 2005; Scholz et al., 2007), and sapropel layer Unit S1 is marked between 3–5.5 m depth (brown shading; Shanahan et al., 2012, 2006).

Broad coherence between data for Hg$_T$, pollen, and ice volume is (perhaps unsurprisingly) clearest following the last glacial termination, which coincides with both the African Humid Period and transition into the Holocene interglacial. Global ice volume is intrinsically linked to North Atlantic sea-surface temperatures (SSTs), and the relationship between these temperatures and West African hydroclimate has been discussed in several studies relevant to the Lake Bosumtwi record (e.g., Gosling et al., 2022b; McKay, 2012; Shanahan et al., 2009), including those relevant to both age models that currently exist for the BOS04-5B core (e.g., Gosling et al., 2022a; Vinnepand et al., 2024). This work has shown that low global ice volumes typically correspond to warmer North Atlantic SSTs and subsequently to moister conditions at Lake Bosumtwi: conditions that appear to also favour higher sedimentary Hg$_T$ values. We agree that the valuable context offered by the LR04 stack should be included more explicitly in our manuscript. We will add these details to the manuscript as follows:

> **Line 195**: "*During the last glacial cycle, moisture availability in West Africa also fluctuated in conjunction with the waxing and waning of high-latitude ice sheets, and their effects on sea-surface temperatures (SSTs) in the North Atlantic (deMenocal, 1995; Weldeab et al., 2007). This teleconnection exists as a function of atmospheric moisture transport and convection processes occurring in the polar and tropical regions. Low global ice volumes typically correspond to warmer North Atlantic SSTs, driving increased atmospheric moisture transport and hence more moist conditions in West Africa. Conversely, high global ice volumes generally correspond to cooler SSTs in the North Atlantic, and subsequently drier conditions in West Africa (e.g., Crocker et al., 2022; Lupien et al., 2023; Stager et al., 2011; Tjallingii et al., 2008).*"

**Line 521**: "*Furthermore, changing sedimentary TOC, terrigenous material, and pollen concentrations all corroborate a broad increase in local moisture availability, temperature, and humidity following deposition of the AI-1 unit (**Fig. 4**): a signal that coincides with the transition into the warmer Holocene interglacial, marked by reduced global ice volume and increased sea surface temperatures in the North Atlantic (McKay, 2012; Scholz et al., 2007; Shanahan et al., 2008b).*"

…and perhaps this should be discussed in the context of the age model. The record should be presented using the newest published version of the core's age model from Vinnepand 2024. This will provide more detail throughout, and despite the lower resolution Hg data around this 73 ka time, may help pin down when this shift is, potentially in the broader context of drivers like ice volume.

While Vinnepand et al. (2024) present the most recent iteration of the BOS04-5B chronology, the resolution of tie-points used in their age-depth model in the interval of this study is very limited (n=2). The BOSMORE7 chronology used in our work utilizes a combination of radiocarbon (calibrated $^{14}$C; n= 109), optically stimulated luminescence (OSL; n=22) and uranium-thorium (U/Th; n=5) dates as independent tie-points (Shanahan et al. 2013). All are concentrated within the youngest section of the core (see figure below). The application of multiple, independent, direct dating techniques to a single core succession continues to provide valuable chronological constraints for a growing number of long, Pleistocene-age sedimentary sequences (e.g., Roberts et al. 2018; Stockhecke et al. 2016). Here, this approach provides age control that is entirely independent from assumptions about past environmental conditions, and unaffected by any age uncertainty stemming from inter-site correlations; such as with the LR04 composite stack (Lisiecki and Raymo, 2005).

The lower resolution of the Vinnepand et al. (2024) chronology creates substantial discrepancies with other age-depth models in the upper ~47 m of the record. This not unsurprising given that the aforementioned study did not include $^{14}$C ages in their model generation, given their aim was to generate an age-depth relationship for a much larger (and thus older) core section (~292 m) than that in our study (~47 m): "*Note that we do not plot $^{14}$C ages (Shanahan et al., 2013) considered by Gosling et al. (2022) as these are not comparable to our study given the temporal application limits of 14C dating.*" (**Fig. 9** caption in Vinnepand et al. 2024; p. 11). However, this smoothing is also problematic. First, because it does not account for the high degree of variability in Lake Bosumtwi's sedimentation regime during the Late Pleistocene (e.g., Shanahan et al. 2013; McKay, 2012). Second, because it is offset from the absolutely dated BOSMORE7 chronology by >10-kyr for sediments >30 m depth. An offset that, in combination with ~10-kyr uncertainties for each of the three tie points in the upper 50 m of the core, substantially reduces the overall precision of the model with respect to our study interval (see **Fig. X** below).

We therefore feel the BOSMORE7 model (Gosling et al. 2022) remains more appropriate, because it explicitly includes absolute dating points and their uncertainties within the here studied interval. Whereas, the Vinnepand et al. model will be a key study for the deeper part of the record where few of those constraints are available. We have illustrated the difference between the two models in **Figure X** below.

[Figure]

**Figure X**: Comparison of age-depth models constructed for the BOS04-5B core by Gosling et al. (2022) (black line), and Vinnepand et al. (2024) (purple line). Panel (a) focusses on the core section relevant to this study, equating to 0 – 47 m of the composite depth. Panel (b) shows this section relative to the full, ~292 m-long BOS04-5B core succession. A black star marks the base of the crater, where $^{40}Ar/^{39}Ar$ dating of shocked quartz yields an age of 1.08±0.04 (Jourdan et al. 2009). Data for the Vinnepand et al. (2024) model are plotted as presented in the supplementary information file presented with the source paper.

Nonetheless, although we retain the chronology as in our original work, we will add reference to the work of Vinnepand et al. (2024) in our manuscript main text:

**Line 223**: "*The upper ~47 m of sediment corresponds to the interval ~96 ka to present, and contains a series of distinct lithological features suggesting pronounced, climate-driven changes in lake level, catchment structure, and sediment transport processes (Vinnepand et al. 2024; Gosling et al., 2022a; McKay, 2012; Miller et al., 2016).*"

**Line 143**: "*These changes all correspond to moisture-driven oscillations between a forest and grass-dominated catchment in response to insolation-driven variability in WAM strength, and migration of the Intertropical Convergence Zone (ITCZ) (e.g., Vinnepand et al. 2024; Gosling et al., 2022a; Miller et al., 2016; Peck et al., 2004)*"

We will also add more explicit justification for use of the BOSMORE7 chronology to our supplementary information file:

**S3. Chronology**

"*…In 2024, Vinnepand and colleagues used cyclicity in total natural gamma ray (NGR) data to create a cyclostratigraphic age-depth model for the full (~946-kyr) BOS04-5B core. This model will be key for study of the deeper (>200 ka) core sections where fewer absolute age markers are available (Shanahan et al. 2013). However, it is significantly lower resolution than the BOSMORE7 chronology in the upper ~47 m of the record, given that Vinnepand et al. (2024) did not include $^{14}C$ ages in their model generation. Not only does this limit the extent to which it can account for the high variability in Lake Bosumtwi's sedimentation regime during the Late Pleistocene (e.g., Shanahan et al. 2013; McKay, 2012), but it also creates a >10-kyr offset from the BOSMORE7 chronology for sediments >30 m depth. An offset that, in combination with ~10-kyr uncertainties for each of the three tie points in the upper 50 m of the core, substantially reduces the overall precision of the model with respect to our study interval. it does create substantial discrepancies. Therefore, given that the BOSMORE7 model explicitly includes absolute dating points and their uncertainties within the here studied interval, it provides age control for our data that is entirely independent from assumptions about past environmental conditions, and unaffected by any age uncertainty stemming from inter-site correlations.*"

**Minor comments:**

I suggest adding a sentence or rephrasing the last sentence of the abstract to have a broader outlook on Hg as a proxy for hydroclimate moving forward. In the same vein, I may suggest starting the

introduction more focused on the importance of reconstructing hydroclimate, rather than on Hg – and the second paragraph of the intro, too, means that this paper is focusing on the Hg proxy development, rather than understanding Bosumtwi hydroclimate. The mix of both directions in this paper is certainly a strength, but the goal of pinning down the Hg proxy is ultimately to understand hydroclimate.

We thank the reviewer for showing great interest in our work, and equally for suggesting the potential for using Hg as a hydroclimate proxy that we could explore with our record. It is an angle we certainly believe is worth pursuing in the future, given how our results show that Hg deposition may include a substantial hydrological component under the right circumstances, such as in Bosumtwi. However, it is uncertain how widely this may be applicable, to what extent Hg may directly reflect hydrology in lake sequences, and/or if hydrological signals are imprinted through certain catchment processes.

- There is a need for better characterization of the processes impacting Hg cycling in lacustrine sediments on millennial timescales, and the ways in which Hg sources, reactions, and transformations within the sedimentary environment could change in time and space (e.g., Frieling et al., 2023; Tisserand et al., 2022; Kovács et al., 2024).
- It is not yet fully clear which factors may pre-determine the sensitivity of a lake system to climate-driven perturbations in Hg cycling. Recent research has shown that geographically and/or structurally similar basins can record different sedimentary Hg signals in response to the same modes of environmental change (e.g., Paine et al., 2024), yet existing long sedimentary Hg records have typically focused on last glacial-age successions (<65 ka; e.g., Sahoo et al., 2023; Wang et al., 2024), or been limited to a single geographical domain (e.g., the Arctic; Gleason et al., 2017).

As a result, in this manuscript, we are cautious to not over interpret the data and present Hg deposition as a proxy for millennial-scale hydro climatic change. We do acknowledge that, provided the hydrological component of the Hg cycle can be isolated, it may present some opportunity to explore this as a proxy for hydrological changes in records such as that obtained from Bosumtwi. For example, measurement of Hg isotopes (e.g., $\delta^{200}$Hg and $\delta^{204}$Hg) in sediments obtained from low-latitude and/or closed lakes could be used to quantify the contribution of Hg to the sediment from precipitation or dry deposition, relative to terrestrial remobilization of Hg stored in plants and soils (Blum et al., 2014; Yin et al., 2024). This would be promising work for a future study.

We will highlight these areas for future investigation in the manuscript in **section 6**:

Line 654: "*Provided the hydrological component of the Hg cycle can be isolated, better characterization of the processes impacting lacustrine Hg cycling could also allow this element to be used as a proxy for hydroclimatic change in terrestrial archives. For example, measurement of sedimentary Hg isotopes in low-latitude and/or closed lakes could quantify the contribution of Hg to the sediment from precipitation or dry deposition, shedding new light on key biogeochemical reaction pathways, processes (e.g., mass-independent fractionation (MIF)), and responses to changing local hydrology across a range of timescales (e.g., Blum et al., 2014; Gao et al., 2023; Yin et al., 2024).*"

**Figure 4**: no need to show precession, and consider including key insolation curves such as 20 or 30 N and perhaps the insolation gradient (23 N – 23 S) and 65 N to test for drivers of hydroclimate throughout the late Pleistocene

We agree that precession is somewhat redundant here, and so will remove this curve from **Figure 4**. However, we also agree that insolation gradient is contextually important to our study as a key factor controlling precipitation distribution in tropical sub-Saharan Africa. With this in mind, we will add three new curves for 6°N (the latitude of Lake Bosumtwi) to **Figure 4**:

**Summer (June) insolation** → higher values typically correspond to wetter summers in tropical sub-Saharan Africa, as higher summer insolation increases surface heating, atmospheric convection, and ITCZ strength.
**Winter (December) insolation** → lower values typically correspond to drier winters, owing to weaker atmospheric convection and reduced precipitation potential.
**Seasonal contrast between June and December insolation** → larger seasonal contrast typically indicates stronger monsoonal dynamics, as the difference in heating between

summer and winter drives the atmospheric pressure gradients that control monsoon circulation. An effect of this would be more pronounced wet seasons.

[Figure]

**Figure 4:** Comparison of key proxy datasets. Included are (from bottom to top), total mercury (Hg_T) and mercury accumulation rate (Hg_AR) for Lake Bosumtwi from this study, chosen as the most appropriate proxies for Hg variability in this core (see **section 5.1**). The first principal component (PC1) of the BOS04-5B XRF data (39% of total variance) is strongly associated with terrigenous elements, and so interpreted as an indicator of lake level changes (McKay, 2012). Forest (woody) taxa abundance (presented as DCA Axis 1; Gosling et al., 2022a; Miller et al., 2016). Lack of data for woody taxa presence is assumed to imply a savannah-dominated regional landscape. Insolation at 6°N (location of Lake Bosumtwi) in June (summer) and December (winter) are calculated following the astronomical solution presented by Laskar et al. (2004) (accessed via. https://vo.imcce.fr/insola/earth/online/earth/online/index.php), and used to calculate seasonal insolation contrast at the Bosumtwi site since ~100 ka.  Also shown is a record of benthic foraminiferal calcite δ18O (‰) derived from the LR04 global stack) as a proxy for ice volume, with cold glacial stages defined by high δ18O ratios (Lisiecki and Raymo, 2005a). Proxy data are all presented on the BOSMORE7 chronology. Unit AI-1 is marked between 33.5 and 32.8 m depth (grey shading; Brooks et al., 2005; Scholz et al., 2007), and sapropel layer Unit S1 is marked between 3–5.5 m depth (brown shading; Shanahan et al., 2012, 2006).

**Line 65**: is it precipitation pattern? Or precipitation amount/strength? Or something else?

We agree the previous phrasing of this sentence was nonspecific. We will address this by amending the sentence as follows:

**Line 69**: "*For example, changes in precipitation __amount__ can influence the proportion of Hg removed from the atmosphere by wet versus dry deposition…*"

**Line 94**: perhaps start with a sentence that is a bit more focused for this study

We agree this sentence (as was previously written) was overly broad. With this in mind, we will remove this sentence so that the paragraph starts as follows:

**Line 97**: "*In the short-term, variability in hydroclimate may manifest as annual changes in rainfall intensity and seasonality, or by sub-decadal fluctuations in regional-scale climate modes (e.g., El-Nino Southern Oscillation, North Atlantic Oscillation; Hernández et al., 2020)."*

**Line 120**: what is its domain?

Well spotted, this sentence does not make full sense. We will amend this to read:

**Lines 122-124**: "*In Sub-Saharan Africa, the West African Monsoon (WAM) regulates precipitation amount and distribution, and drives long-term evolution of environmental characteristics and mineral-dust emissions…"*

**Line 122**: perhaps add some more citations for orbital control of the WAM seen by leaf waxes, like from O'Mara and Kuechler

This is a good suggestion. Given the relevance of these studies to the point made here, they will be incorporated into the manuscript as follows:

**Lines 122-125**: *"In sub-Saharan Africa, the West African Monsoon (WAM) regulates precipitation amount and distribution, and drives long-term evolution of environmental characteristics and mineral-dust emissions (O'Mara et al. 2022; Kaboth-Bahr et al., 2021; Kuechler et al. 2013; Weldeab et al., 2007)."*

**Line 124**: I suggest changing drought events to arid periods, if these records are focused on orbital-scale climate variability

Good point, and we thank reviewer #1 for pointing out where different terminology is required. We will adjust this sentence to read:

**Lines 125-128**: "*Proxy records from this domain show that orbitally-driven variations in the strength of the WAM have frequently driven distinct arid (Cohen et al., 2007; Scholz et al., 2007) and humid periods (Armstrong et al., 2023; Menviel et al., 2021) throughout the Pleistocene."*

**Line 129**: over what time period?

We agree more specificity is needed here. Sentence will be amended to read:

**Lines 133-136**: "*Here our focus is on sediment core BOS04-5B extracted from Lake Bosumtwi, Ghana (West Africa): a core that provides a clear and continuous record of this hydroclimate variability covering the late Pleistocene…"*

**Line 161**: maybe add one sentence or phrase about how this makes the Lake Bosumtwi archive ideal for refining the Hg-paleoclimate method or generating paleoclimate reconstructions in general

We agree this is key knowledge to include and indeed discussed this specifically in our original manuscript in two dedicated paragraphs:

(1) In **section 1.2**, where we outline our research objectives and thus highlight the suitability of Lake Bosumtwi for this purpose

(2) In **section 2**, where we dedicate a full section that outlines why this sedimentary record is well suited to fulfil our research objectives:

**2.1.3.    Paleoclimatic significance**

We will carefully revisit these during revisions to ensure all information is included, easily accessible, and add a specific reference to studying Hg behaviour. For example, the addition of the following sentence to our introduction:

**Lines 170-174**: "*These properties render the lake highly sensitive to changes in atmospheric processes, but also imply that Hg inputs may originate exclusively from the atmosphere (e.g., by wet deposition). Therefore, this system is ideally suited for exploring whether specific basin characteristics (e.g., depth, nutrient status, bathymetry) could also measurably affect how Hg signals are encoded in the sedimentary record…"*

**Line 163**: perhaps just leave out this short-term definition, as only seasonal changes are described. Unless these vary on 10 year timescales? The description goes from less than 10 to over 1000 year variability.

> Fair point, we will remove the short-term definition so the sentence reads:
>
> **Line 177:** "*Seasonal variability in the tropical rain belt position drives short-term hydroclimate change in West Africa.*"

**Section 2.1.1**: is the ITCZ also expanding and contracting? Rather than just moving and strengthening/weakening? Also, there is more evidence that the insolation gradient, rather than just precession, is controlling the WAM during the Pleistocene (see O'Mara et al 2024)

> Both great points, and we agree that the complexities associated with West African climate during the Pleistocene should be given more detail in this section. To better describe this complexity, we will edit the following sentences to read:
>
> **Lines 185-187**: "*Changes in axial precession produce fluctuations in seasonal insolation above the African continent, influencing the strength of the WAM, the Walker Circulation, the position and dimensions of the ITCZ, and the availability of continental moisture…*"
>
> **Lines 188-191**: "*Several studies have shown weakening of the WAM and southward migration of the ITCZ in response to high precession and/or changes in insolation gradient, producing drier conditions in West Africa and subsequent reductions in terrestrial precipitation, ecosystem productivity, and recession of terrestrial forests (O'Mara et al. 2022; Menviel et al., 2021).*"
>
> Reviewer 1 also makes a great point regarding the importance of insolation, and we will directly respond to this by adding insolation curves for summer (June), winter (December), and overall seasonal contrast at 6°N (the latitude of Lake Bosumtwi) to **Figure 4** (*see below*). Together, these three curves will provide a more comprehensive picture of insolation variability at our study site during the Late Pleistocene, and so give us opportunity to underscore the importance of orbital forcing for variability in precipitation over long timescales in this domain.

[Figure]

**Figure 4:** Comparison of key proxy datasets. Included are (from bottom to top), total mercury (Hg$_T$) and mercury accumulation rate (Hg$_{AR}$) for Lake Bosumtwi from this study, chosen as the most appropriate proxies for Hg variability in this core (see **section 5.1**). The first principal component (PC1) of the BOS04-5B XRF data (39% of total variance) is strongly associated with terrigenous elements, and so interpreted as an indicator of lake level changes (McKay, 2012). Forest (woody) taxa abundance (presented as DCA Axis 1; Gosling et al., 2022a; Miller et al., 2016). Lack of data for woody taxa presence is assumed to imply a savannah-dominated regional landscape. Insolation at 6°N (location of Lake Bosumtwi) in June (summer) and December (winter) are calculated following the astronomical solution presented by Laskar et al. (2004) (accessed via. https://vo.imcce.fr/insola/earth/online/earth/online/index.php), and used to calculate seasonal insolation contrast at the Bosumtwi site since ~100 ka. Also shown is a record of benthic foraminiferal calcite δ$^{18}$O (‰) derived from the LR04 global stack) as a proxy for ice volume, with cold glacial stages defined by high δ$^{18}$O ratios (Lisiecki and Raymo, 2005a). Proxy data are all presented on the BOSMORE7 chronology. Unit AI-1 is marked between 33.5 and 32.8 m depth (grey shading; Brooks et al., 2005; Scholz et al., 2007), and sapropel layer Unit S1 is marked between 3–5.5 m depth (brown shading; Shanahan et al., 2012, 2006).

**Lines 221-224**: Nobody will disagree with this statement, but an additional clause indicating why lake level alone is not a perfect measure of precipitation will lead well into the study of Hg.

> Good suggestion. We will add the suggested clause onto the end of this paragraph as follows:

> **Lines 244-245**: "*The closed hydrology of Lake Bosumtwi means that changing water levels will primarily reflect the magnitude of precipitation variability in the region, with higher lake levels typically occurring during wetter climate intervals (Russell et al., 2003; Shanahan et al., 2008b). However, lake level may have also been influenced by secondary processes such as evaporation, and, over long time-scales, sediment infill (McKay, 2012).*"

I recommend reporting correlation as r, rather than r$^2$, so we can see numerically the direction of correlation.

**Figure 3**: as mentioned above, if presented as r rather than r$^2$, there won't be a need for the italics and it'll be clearer.

We concur that presenting r values (in addition to r²) provides useful additional information regarding the relationships between the variables. To display this information more clearly, we will add r-values to the scatter plots presented in **Figure 3**, alongside the r² values:

[Figure]

**Figure 3: (b)** Comparison of relationships in Lake Bosumtwi between ~96 and 73 ka (black circles), and between ~73 and 0 ka (stars). We first assess the $Hg_T$ record for this lake relative to two potential host-phases: total organic carbon (TOC) values measured in this study, and detrital minerals (estimated by potassium (K)) concentrations measured by McKay (2012). We also test the relationship between $Hg_T$ and first principal component (PC1) of the BOS04-5B XRF data, in which 39% of total variance is associated with terrigenous elements. PC1 was consequently interpreted as an indicator of lake level changes (McKay, 2012). R (r) and r-squared (r²) values for each interval are also given. The significance of all correlations were assessed using a Student's t-test, which showed that all three combinations were significant at p<0.01. Stars marked in teal correspond to deposition of sapropel unit 1 (S1) in BOS04-5B.

And when correlations are presented, please include number (n) and significance (p).

We agree this information is critical for presentation of our results, and thank the reviewer for pointing this out. This also aligns with a comment made by reviewer #2. With these comments in mind, to present statistical information in sufficient detail, and ensure our results are fully transparent, we will make the following amendments:

- Addition of p-value information to the caption and main body of **Figure 3** (*see above*)
- Addition of both r and r² values to relevant points in the manuscript. For example:

    **Lines 432-437**: *"An overall positive association between $Hg_T$ and TOC (r = 0.64; r² = 0.42) suggests that Hg variability may be associated with organic carbon variability in Lake Bosumtwi. However, it is noteworthy that detrital materials (e.g., K) show negative correlations with both TOC (r = -0.73; r² = 0.53) and Hg (r = -0.60; r² = 0.34) so that the Hg-TOC correlation may reflect, in part, a correlation imposed by variable clay-dilution of both Hg and TOC. Moreover, these correlations are all significant at p <0.001 (unless stated otherwise)."*

    **Line 447**: *"A negative overall correlation between $Hg_T$ and K is apparent throughout the record (r = -0.60; r² = 0.34; **Fig. 3b**)."*

- Addition of a new sheet to our supplementary data file, which will contain three tables: (1) r values, (2) t-squared statistics, and (3) p-values for all for all geochemical combinations analysed, for which it is not practical to include in the figure or text. This sheet is titled: "**Pearson's Correlation**"
- Addition of a new section to our supplementary file that provides a more detailed description of the correlation analyses applied to BOS04-5B:

    **S8. Correlation analyses**

- Amendment of the **Figure 3a** caption to include values for (n) and (p):

    *"Full-core correlation (Pearson's r) matrix for Hg, total organic carbon (TOC) (this study), and a suite*

*of trace elements measured in BOS04-5B by XRF (McKay, 2012). Higher r values suggest that similar processes influence the concentration of the two elements in focus). Sample size (n) was 157 for each analysis, and ~75% of the assessed trace element combinations were significant (p<0.01). The 25% that were not are marked with an asterisk (\*)."*

**Line 517**: some records show a drying trend towards the LGM, particularly in eastern Africa (Garelick, Baxter, Lupien, Tierney). Please discuss how this fits into your discussion, if the focus is as trans-continental as it is now.

Interesting observation. We agree that the "*broader, regional-scale shift in hydroclimate across sub-Saharan Africa*" we mention in our discussion should have been explained in a bit more detailed context. We also thank reviewer #1 for referring us to key papers in this context. With these comments in mind, we will amend the paragraph to read:

**Lines 566-578**: "*Proxy data generated from the BOS04-5B core suggest that progressively wetter conditions affected the catchment following ~73 ka (e.g., Gosling et al., 2022a; Shanahan et al., 2008b). Paleoclimate records based on the sediments of lakes Malawi (Tanzania), Bambili (Cameroon), Tanganyika (Tanzania/Democratic Republic of the Congo), Chew Bahir (Ethiopia) and Chala (Tanzania) (e.g., Cohen et al., 2007; Foerster et al., 2022; Lézine et al., 2019; Scholz et al., 2007), and marine sediment core material from the West African margin (Figs. 1, 5) (e.g., Kinsley et al., 2022; Skonieczny et al., 2019) also pertain to a distinct regional-scale shift in hydro climate across tropical sub-Saharan Africa at this point in time (Fig. 5). Specifically, a shift characterized by a distinct moisture gradient favouring wetter conditions in the west of the continent relative to the east, which was further amplified during the last glacial termination (e.g., Gosling et al. 2022a; Baxter et al. 2023; Lupien et al. 2022). Therefore, the coeval increase in the frequency and amplitude of Hg enrichments in Lake Bosumtwi, and associated rises in lake level, could indirectly reflect pronounced shifts in hydroclimate across tropical sub-Saharan Africa.*"

**Lines 526-527**: this is a narrow statement – see a nice explainer of fuel-limited and moisture-limited by Karp 2023

We agree this statement is overly simple. We will re-write this sentence as follows and include the recommended citation:

**Lines 579-589**: "*Hydroclimate was a key driver of changes in fire activity in tropical sub-Saharan Africa during the late Pleistocene. However, although wetter climatic conditions may be broadly associated with heightened fire activity due to associated increases in terrestrial biomass, recent work has shown that discrete changes in precipitation can elicit notably different fire responses between sites (e.g., Karp et al. 2023; Gosling et al., 2021; Moore et al., 2022) . The influence of biomass burning on the Hg record presented here appears similarly complex; despite being a well-constrained factor in the Bosumtwi catchment, and evidence that wildfires are also a significant source of Hg, accounting for ~13% of natural Hg (re-)emissions to the modern atmosphere (Francisco López et al., 2022). Given that no clear relation is visible between $Hg_T$, $Hg_{AR}$, and two discrete macro- (Kiely, 2023) and micro- (Miller et al., 2016) charcoal profiles generated from the BOS04-5B core, we suggest that the effects of Hg emitted during wildfires did not leave a clear imprint on Hg variability in this record (Fig. S9).*"

**Technical**:

**Lines 63-64**: remove one of the 'direct's

These two sentences will be edited to read:

**Line 67**: "*The transport and transformation of Hg at the Earth's surface is linked to the hydrological cycle (Bishop et al., 2020; Selin, 2009). Water plays a direct role in the efficiency of both Hg deposition and re-emission.*"

**Line 148**: change Ma to Myr-old, or "dated to 1.08 Ma"

Sentence will be edited to read:

**Line 154**: "*It occupies a meteorite impact crater dated to 1.08 ± 0.04 Ma, which is one of the youngest and best preserved impact craters on Earth…*"

**Line 230**: cite the map in Fig 1

Good suggestion. Reference to **Figure 1** will be added as follows:

**Line 254**: "*Our study focuses on the upper ~47 m section of a 296-m-long core extracted from deep-water (76 m) site 5 (core BOS04-5B; **Fig. 1**)…*"

**Line 240**: changed generated to described or similar

We will edit wording of this sentence to read:

**Line 273**: "*Age control for the ~47 m of sediment analysed in this study is provided by the BOSMORE7 model, presented by Gosling et al. (2022).*"

**Line 256**: might help to include the maximum spacing between samples too

Agreed. This sentence will be amended to:

**Lines 288-290**: "*For this study, we analysed 165 samples spanning the composite depth interval 47.7 to 0 m, with an average temporal resolution of ~0.6 ka between each sample (range: 0.01 to 5.85 kyr).*"

**Line 261-262**: can simply cite the supplemental data, rather than including a sentence

The full sentence will be deleted, and we will amend the previous sentence as follows:

**Line 295**: *"and then one standard for every 10 lacustrine samples (supplementary data)."*

**Lines 383-385**: instead of talking directly about Figure 2, try removing the first part of this sentence and then citing Figure 2 at the end.

We agree this longer reference to Figure 2 adds unnecessary bulk to the sentence. From this, the sentence will be edited as follows:

**Line 429:** "*Two mechanisms emerge as plausible drivers of Hg variability in Lake Bosumtwi (**Fig. 2**).*"

**Line 386**: I don't think this last sentence of the paragraph is necessary – there are a lot of these statements throughout that are a bit redundant that can be cut down

We agree with this point, and so will amend this paragraph so that the sentence in question is removed:

**Lines 426-430**: "*Studying time-resolved changes in lake sediment Hg concentration provides a valuable opportunity to study changes in the pre-industrial Hg cycle, how these changes translate to measurable sedimentary signals, and their links to local and regional-scale environmental variability (Cooke et al., 2020). Two mechanisms emerge as plausible drivers of Hg variability in Lake Bosumtwi (**Fig. 2**). First is organic matter (host) availability, and second is external change in net Hg input to the system.*"

We will also carefully revisit the entire text to eliminate any other superfluous statements.

**Line 516-517**: "these records" needs clarification aside from the figure citation.

The section being referenced here will be amended so to read:

**Lines 568-573**: "*Paleoclimate records produced from the sediments of lakes Malawi (Tanzania), Bambili (Cameroon), Tanganyika (Tanzania/Democratic Republic of the Congo), Chew Bahir (Ethiopia) and Chala (Tanzania) (e.g., Cohen et al., 2007; Foerster et al., 2022; Lézine et al., 2019; Scholz et al., 2007), and marine cores extracted from the West African margin (**Figs. 1, 5**) (e.g., Kinsley et al., 2022; Skonieczny et al., 2019) also pertain to a distinct regional-scale shift in hydro climate across sub-Saharan Africa at this point in time (**Fig. 5**).*"

**Line 526**: citation

We will make the following amendment to this opening statement:

**Lines 579-580**: "*Hydroclimate was a key driver of changes in fire activity in sub-Saharan Africa during the late Pleistocene (Moore et al. 2022).*"

**Sections 5.2.1 and 5.2.2**: I suggest renaming these sections into more descriptive titles

We agree that the original section headers are overly vague, and would benefit from being more descriptive. With this in mind, we will adjust the section headers to read:

**5.1.1. Arid conditions (~96 to 73 ka)**

**5.1.2. Humid conditions (~73 to 0 ka)**

**Line 539**: change ka to kyr

The text will be corrected to read:

**Line 594**: "*The resolution of the BOS04-5B record (~0.6 kyr per sample)…*"
* * *
**REVIEWER #2**

This manuscript uses a sediment core collected that dates to 96k to link Hg to climatic events in West Africa. The manuscript is well written, and the methods and QA/QC are clearly explained.

We give earnest thanks to reviewer #2 for their kind and constructive feedback on our manuscript. In the response below and in our revised manuscript, we will endeavor to ensure the edit suggestions are considered and addressed, with alterations made where necessary.

I would remove the word new from the title.

Fair suggestion, we agree this word is superfluous and so will remove it from the main manuscript and the supplementary file. Our manuscript will subsequently be titled:

"**Evidence for millennial-scale interactions between Hg cycling and hydroclimate from Lake Bosumtwi, Ghana**"

There do not appear to be any keywords. These should not be the same as words in your title.

The following keywords will be added to the manuscript below the abstract:

**Line 36**: "***KEYWORDS***: *lacustrine, mercury, geochemistry, sediment, sapropel, Lake Bosumtwi, Africa*"

**Line 111**: There is a colon in this line that should not be there.

The colon will be removed so the sentence reads:

**Lines 112-115**: "*However, few terrestrial Hg records extend beyond the present interglacial (>12 ka), and even fewer come from the low-latitudes, where tropical rainforest, grassland and desert biomes are highly sensitive to millennial-scale hydroclimate variability (Bradley and Diaz, 2021; Schneider et al., 2023)*"

**Line 128**: focusses is not correct use focus

We agree the grammar of this sentence could be improved and we will revise this sentence to read:

**Lines 134-136**: "*Here our focus is on sediment core BOS04-5B extracted from Lake Bosumtwi, Ghana (West Africa): a core that provides a clear and continuous record of this hydroclimate variability covering the late Pleistocene…*"

The section 1.3 titled West African Monsoon should be changed from this section basically introduces the reader to the research objectives. Do you have a research hypothesis?

We thank the reviewer for highlighting this discrepancy. Considering this comment, we have revisited how to best structure the introduction and made some adjustments, particularly in section headers to improve the flow and better guide the reader to our research objectives. For example, we like the sub-title suggestion made by reviewer 1 here, and so the title of this section will be edited to now read:

**Line 120**: "*1.2. Research objectives*"

Further, we will carefully revisit the introduction so the research hypothesis – **changes in hydroclimate can affect how mercury (Hg) is transported and buried in terrestrial sediments** – is clearly presented. This hypothesis was purposefully broad in order to provide a solid baseline for assessing the presence (or absence) of any Hg-climate relationships in the Bosumtwi record using both observational and statistical methods; while also accounting for the fact that a widely accepted framework for how the terrestrial mercury (Hg) cycle responds to millennial-scale climate changes, and how these responses vary by location, is still lacking. To this end, we will also incorporate the following statement into our revised manuscript:

**Lines 130-133:** *"In light of growing evidence for a hydroclimatic influence on the terrestrial Hg cycle (e.g., Guédron et al., 2018; Nalbant et al., 2023; Paine et al., 2024), we hypothesized that humid and/or arid periods in sub-Saharan Africa would have elicited measurable changes in the Hg cycle, producing measurable signals in regional sedimentary records."*

**Line 139**: I would get rid of aims to assess and just use "assessed"

This sentence will be revised to read:

**Lines 144-146**: *"Focussing on the uppermost ~47 m of the Lake Bosumtwi sediment record, this study assesses whether major shifts in local hydroclimate produced measurable changes in how Hg has been transported to, and buried within, this system since ~96 ka"*

**Line 142**: explore should be explored. Note: I am a strong believer that what was done should be described in past tense.

Well spotted. Sentence will be amended to read:

**Lines 147-149**: "*By comparing our sedimentary Hg record with proxy data from archives across the African continent (e.g., Foerster et al., 2022; Scholz et al., 2007), we explored whether hydroclimate has exerted a measurable effect on terrestrial Hg cycling in the WAM domain in over the past ~100-kyr.*"

WAM and ITCZ should be written out the first time used with the abbreviation afterwards in parenthesis.

Agreed, and we will ensure we present these terms with the appropriate abbreviations the first time they are used in the manuscript:

**Line 122:** "*…the West African Monsoon (WAM)…*"
**Line 143:** "*…Intertropical Convergence Zone (ITCZ)*"

**Line 217**: which should be that, and there should be no comma preceding this

Sentence will be amended to:

**Lines 238-240:** "*The core also shows distinct co-enrichment in manganese (Mn) and iron (Fe) in certain intervals following AI-1, that are associated with manganosiderite (Mn-rich $FeCO_3$) precipitation in the lake sediments*"

**Line 230**: which should be that. Which is used when indicating something general, while that is used when discussing something specific.

We agree this is the grammatically correct form, and so should have been used here. We will correct this sentence to read:

**Lines 254-256:** "*Our study focuses on the upper ~47 m section of a 296-m-long core extracted from deep-water (76 m) site 5 (core BOS04-5B), that extends from the present-day lake floor to the brecciated bedrock dated by $^{40}Ar/^{39}Ar$ to 1.08±0.04 Ma (Jourdan et al., 2009)*"

**Line 260**: run should be analyzed

> Agreed. Our wording will be corrected:
>
> **Lines 294-295:** "…*with a known Hg value of 290 ± 9 ng g$^{-1}$ were analysed to calibrate the instrument before sample analysis, and then one standard for every 10 lacustrine samples*"

Please explain in the methods how the core was stored and sampled. It is very easy to contaminate samples with Hg. This must be clearly addressed.

> We agree details on core handling and storage are useful to include and will add the following details to the manuscript in section **3.1. BOS04-5B**:
>
> **Lines 262-270**: "*After drilling in 2004, core BOS04-5B was shipped to the University of Rhode Island and split. The physical properties of the full ~296 m core were measured at 2-cm intervals using a Geotek ® multi-sensor core logger (Koeberl et al., 2007). After logging and imaging and at 4-cm resolution, 2 cm thick slices were removed from the core half and separated into sub-samples for multi-proxy analyses, including sediment magnetic hysteresis, x-ray diffraction mineralogy, total organic and inorganic carbon content, bulk organic carbon and nitrogen isotopes, grain size, pollen, and charcoal (e.g., Gosling et al., 2022; McKay, 2012; Miller et al., 2016). Following bulk sediment analyses, the BOS04-5B core material was transferred to the Continental Scientific Drilling (CSD) Repository in Minneapolis.*"
>
> We also more clearly refer the reader to additional details related to the ICDP coring operation during which core BOS04-5B was extracted which are included in our supplementary information, under the section:

> ### S1. BOS04-5B core details

**Line 276**: which should be that. I am not going to mention any more of these and will leave this up to the authors to correct.

> We thank reviewer #2 for flagging this grammatical error, and have carefully checked for other instances throughout the manuscript. The sentence in focus will be corrected to:
>
> **Line 309**: "*Both aliquots were then combusted in oxygen at 1220ºC to break down the calcium carbonate and produce carbon dioxide ($CO_2$), that was fed into a solution of barium perchlorate.*"
>
> We will also go through the full manuscript, and amend **14** instances where we identified this error was present.

**Line 364**: "broadly track" is subjective. It would help if you did some statistical analyses to make this statement more quantitative. Same with the discussion line 365.
**Line 369**: some statistical analyses would help quantify the discussion.
**Line 389**: ah here is the statistical analyses result. Please move up to the results section. Note: r$^2$ values are meaningless without a p-value. Is it worth putting in the equation for the correlation? This might be of interest to others. Don't forget to put your statistical analyses method in the methods section.

> In response to the three comments above: we agree that **section 5.1** (as-was) should be in the results section of the manuscript. Inspired by this comment, we have moved this section such that it is now:

> #### 4.1. Lacustrine host phases of mercury

> With this change, the correlation matrix and its associated information will also be moved to the results section.

> We also agree that statistical analyses are a powerful tool for ensuring the statements made in our discussion are based on quantitative evidence. Equally, we agree that more details of our

statistical analyses need to be included in the methods section of our manuscript, and this also aligns with a comment made by reviewer #1. In light of this suggestion and to present statistical information in sufficient detail, we will make the following amendments:

- All correlation analyses in text or figures are now accompanied by r, $r^2$, and p-values. For example:

    **Lines 432-437**: "*An overall positive association between $Hg_T$ and TOC (r = 0.64; $r^2$ = 0.42) suggests that Hg variability may be associated with organic carbon variability in Lake Bosumtwi. However, it is noteworthy that detrital materials (e.g., K) show negative correlations with both TOC (r = -0.73; $r^2$ = 0.53) and Hg (r = -0.60; $r^2$ = 0.34) so that the Hg-TOC correlation may reflect, in part, a correlation imposed by variable clay-dilution of both Hg and TOC. Moreover, these correlations are all significant at p <0.001 (unless stated otherwise).*"

    **Figure 3**: *(a) Full-core correlation (Pearson's r) matrix for Hg, total organic carbon (TOC) (this study), and a suite of trace elements measured in BOS04-5B by XRF (McKay, 2012). Higher r values suggest that similar processes influence the concentration of the two elements in focus).* **Sample size (n) was 157 for each analysis, and ~75% of the assessed trace element combinations were significant (p<0.01). The 25% that were not are marked with an asterisk (\*).** *Grey shading marks positive correlations (light: >0.25, dark: >0.5), and orange shading marks negative correlations (light: <-0.25, dark: <-0.5). Unshaded boxes mark weak/negligible correlations (between 0 and 0.25, and 0 and -0.25), with values greyed-out for clarity. All remaining values are presented with black text, with those in this range related to Hg in the boldest type. (b) Comparison of relationships in Lake Bosumtwi between ~96 and 73 ka (black circles), and between ~73 and 0 ka (stars). We first assess the $Hg_T$ record for this lake relative to two potential host-phases: total organic carbon (TOC) values measured in this study, and detrital minerals (estimated by potassium (K)) concentrations measured by McKay (2012). We also test the relationship between $Hg_T$ and first principal component (PC1) of the BOS04-5B XRF data, in which 39% of total variance is strongly associated with terrigenous elements, and so interpreted as an indicator of lake level changes (McKay, 2012).* **R (r) and r-squared ($r^2$) values for each interval are also given. The significance of all correlations were assessed using a Student's t-test, which showed that all three combinations were significant at p<0.01.** *Stars marked in teal correspond to deposition of sapropel unit 1 (S1) in BOS04-5B*"

    The only exception to this is the correlation matrix (**Fig. 3**) where we supply these numbers in a data sheet (see following point).
- Addition of a new sheet to our supplementary data file, which will contain three tables: (1) r values, (2) t-squared statistics, and (3) p-values for all for all combinations analysed. This sheet is titled: "**Pearson's Correlation**"
- Addition of a new section to our supplementary file that provides a more detailed description of the correlation analyses applied to BOS04-5B:

    **S8. Correlation analyses**

- Amendment of the **Figure 3** caption to include values for (n) and (p):

    "*Full-core correlation (Pearson's r) matrix for Hg, total organic carbon (TOC) (this study), and a suite of trace elements measured in BOS04-5B by XRF (McKay, 2012). Higher r values suggest that similar processes influence the concentration of the two elements in focus).* **Sample size (n) was 157 for each analysis, and ~75% of the assessed trace element combinations were significant (p<0.01).**"

Are there any Hg analyses of the crater walls? I think it would be a good idea to add a description of the geology associated with the crater area in the site description.

To the best of our knowledge there are unfortunately no direct Hg analyses available for the material surrounding the lake. We concur that a more detailed description of the local geological setting should have been included in our site description. Further details will be added in the main text:

**Line 153-158**: "*Lake Bosumtwi is the only natural lake in Ghana, West Africa (6°30' N, 1°25' W) (Fig. 1). It occupies a 1.08 ± 0.04 Ma meteorite impact crater, which is one of the youngest and best preserved impact craters on Earth (Jourdan et al., 2009). The surrounding bedrock and meteorite impact target rocks are Proterozoic metagraywackes, phyllites and metavolcanic rocks of the Birimian Supergroup (~2*

*Ga) (Jones et al., 1981). Lake beds, soils, and breccias constitute the most recent rock formations at the site, and are associated with evolution of the crater through time (Koerberl et al. 2007)."*

Although the extent to which the geogenic Hg pool may contribute to signals recorded in Lake Bosumtwi is difficult to constrain, for example due to lack of Hg analyses on the detrital source material, we do explore plausible hypotheses related to this topic in our supplementary information file under the header:

**S12. Crater origin and geology**

In this file, we also include a geological map of the Bosumtwi crater:

[Figure]

**Figure S6:** A schematic geological map of the Bosumtwi impact structure, Ghana. Adapted from Koeberl et al. (2007)

**Figure 3** is blurry.

We apologise for any difficulties reviewer #2 may have had in viewing our manuscript figures. Although no blur is detectable on the PDF file that was submitted to *Climate of the Past* in our original submission, we will make sure to perform additional checks on the manuscript file when uploading our revised submission.

Please identify the p-values associated with the $r^2$ values.

We agree this information should have been included in our original submission, and are grateful to reviewer #2 for flagging this. As mentioned above, we will make this information available in our revised submission by addition of a new sheet to our supplementary data file (titled: "**Pearson's Correlation**"), which will contain three tables: (1) r values, (2) t-squared statistics, and (3) p-values for all for all combinations analysed. We will also include a section in our supplementary file that provides details on how these p-values were calculated, and an additional statement in the caption for **Figure 3:**

*Figure 3: "(a) Full-core correlation (Pearson's r) matrix for Hg, total organic carbon (TOC) (this study), and a suite of trace elements measured in BOS04-5B by XRF (McKay, 2012). Higher r values suggest that similar processes influence the concentration of the two elements in focus). **Sample size (n) was 157 for each analysis, and ~75% of the assessed trace element combinations were significant (p<0.01).** The 25% that were not are marked with an asterisk (\*). Grey shading marks positive correlations (light: >0.25, dark: >0.5), and orange shading marks negative correlations (light: <-0.25, dark: <-0.5). Unshaded boxes mark weak/negligible correlations (between 0 and 0.25, and 0 and -0.25), with values greyed-out for clarity. All remaining values are presented with black text, with those in this range related to Hg in the boldest type. (b) Comparison of relationships in Lake Bosumtwi between ~96 and 73 ka (black circles), and between ~73 and 0 ka (stars). We first assess the $Hg_T$ record for this lake relative to two potential host-phases: total organic carbon (TOC) values measured in this study, and detrital minerals (estimated by potassium (K)) concentrations measured by McKay (2012). We also test the relationship between $Hg_T$ and first principal*

*component (PC1) of the BOS04-5B XRF data, in which 39% of total variance is strongly associated with terrigenous elements, and so interpreted as an indicator of lake level changes (McKay, 2012). R (r) and r-squared (r²) values for each interval are also given. **The significance of all correlations were assessed using a Student's t-test, which showed that all three combinations were significant at p<0.01**. Stars marked in teal correspond to deposition of sapropel unit 1 (S1) in BOS04-5B."*

Can you do some regression analyses for data presented in **Figure 4**? Would this make your discussion more robust?

This is a very good point. In **Figure 4**, we point out that a large number of elements measured in BOS04-5B by XRF are presented by means of PC1. Specifically, those that are strongly associated with the terrigenous elements Al, Si, K, Ti and Rb, whose concentrations are strongly driven by changes in the terrigenous content of core BOS04-5B, the overall sediment supply, and terrigenous drainage area: all functions of lake level (McKay, 2012). Given the importance of lake level for interpretation of our Hg record, we fully agree with reviewer #2 that a quantitative analysis of the relationship between $Hg_T$ and PC1 would be a valuable addition to our revised manuscript. Given the differences in resolution between the two datasets, PC1 values were averaged to obtain a value corresponding to the interval covered by each discrete $Hg_T$ sample (~0.5 cm). We will present the result of this analysis in **Figure 3** in our revised manuscript (*see below*), and refer explicitly to the statistical values in the discussion text. For example:

**Lines 516-521:** "*The magnitude and frequency of variability in $Hg_T$ visibly increases at ~73 (±5) ka (**Fig. 4**). The quantitative significance of this shift is supported by changepoint analysis of the BOS04-5B dataset, which demonstrates a clear and step-wise shift in $Hg_T$ values between ~75 and 73 ka (**Fig. S3**). It also occurs in conjunction with an increase in the lake's water level (**Fig. 4b**), which is corroborated by a statistically-significant relationship between PC1 (lake level indicator; McKay, 2012), and $Hg_T$ in our record (**Fig. 3b** - r = -0.53; r² = 0.29).*"

[Figure]

**Figure 3: (b)** Comparison of relationships in Lake Bosumtwi between ~96 and 73 ka (black circles), and between ~73 and 0 ka (stars). We first assess the $Hg_T$ record for this lake relative to two potential host-phases: total organic carbon (TOC) values measured in this study, and detrital minerals (estimated by potassium (K)) concentrations measured by McKay (2012). We also test the relationship between $Hg_T$ and first principal component (PC1) of the BOS04-5B XRF data, in which 39% of total variance is strongly associated with terrigenous elements, and so interpreted as an indicator of lake level changes (McKay, 2012). R (r) and r-squared (r²) values for each interval are also given. The significance of all correlations were assessed using a Student's t-test, which showed that all three combinations were significant at p<0.01 Stars marked in teal correspond to deposition of sapropel unit 1 (S1) in BOS04-5B.

This same approach could not be taken to apply regression analyses of relationships between the BOS04-5B Hg and forest (woody) taxa abundance (presented as DCA Axis 1), due to differences in sampling position, which would result in significant interpolation. Although the pollen and Hg concentrations were measured in the same core, they were each measured on different sample sets at a similar resolution; meaning that values are offset from each other by

~20 cm (equating to >100 years). This implies interpolation is particularly risky when considering the effects of short-lived events such as wildfires on Hg concentrations, charcoal and pollen, and thus we cannot appropriately test for a relation between these proxies.

Despite these limitations, we fully agree with reviewer #2's comment that exploration of Hg-vegetation-catchment interactions is an important area for future study. Namely, that studies of this nature could provide crucial context for understanding which mechanisms are most significant for transport, accumulation, and cycling of Hg in different ecosystems, and comparing similarities and differences in Hg signals between lakes with comparable or opposing structures (e.g., closed versus open), composition (e.g., grassland versus forest), and sensitivity to hydroclimate.

**Line 501**: please include the Outridge reference since he was the first to state this.

We agree this reference is important to include here, and so will amend the citation to read:

**Lines 548-551**: "*Scavenging of Hg from the water column by algae is also a process now recognised as an important driver of Hg export to lacustrine sediments; particularly in systems where primary productivity, organic matter production, and burial capacity is high (Outridge et al. 2007; Biester et al., 2018; Schütze et al., 2021)*"

**Line 508**: can you provide information on documented fire events in this area that would directly impact water in the crater?

We are grateful to reviewer #2 for highlighting this, as it underscores how we have (wrongly) omitted to mention the existing literature that is relevant here. Namely, studies that have analysed micro charcoal concentrations in the BOS04-5B core, and subsequently used this data to provide millennial scale fire histories for this region. For example:

> Miller, C.S., Gosling, W.D. (2014) Quaternary forest associations in lowland tropical West Africa. *Quaternary Science Reviews* **84**: 7–25

> Miller, C.S., et al. (2016) Drivers of ecosystem and climate change in tropical West Africa over the past ~540000 years. *Journal of Quaternary Science* **31**: 671–677

> Gosling, W.D. et al. (2021) Preliminary evidence for green, brown and black worlds in tropical western Africa during the Middle and Late Pleistocene. *Palaeoecology of Africa* **35**: 13–25

To acknowledge these studies, we will amend this section of our manuscript to read:

**Lines 558-562**: "*For Lake Bosumtwi these direct inputs may have come from precipitation, and/or from increased flux of charcoal (and associated released of Hg from vegetation during wildfires) into the lake following local wildfire events; the latter documented by variations in the micro charcoal concentration of BOS04-5B (Gosling et al., 2021; Miller et al., 2016; Miller and Gosling, 2014).*"

However, it is important to note is that, similar to the pollen data, we could not conduct regression analyses of relationships between the BOS04-5B Hg and charcoal abundance due to differences in sampling position. The BOS04-5B has been assessed for both macro and microcharcoal abundance (Kiely et al., 2025; Miller et al., 2016), however, almost all samples assessed for this purpose were measured on different sample sets at a similar resolution. This creates a ~20 cm (>100 years) offset, and hence reduces the suitability of interpolation for alignment of these datasets; particularly when assessing the effects of short-lived events such as wildfires on Hg concentrations. We cannot entirely discount the possibility that wildfires did affect Hg fluxes into Lake Bosumtwi, although other drivers were likely more influential; namely moisture-driven changes in sedimentation, productivity, and detrital material supply. Details of this analysis will be provided in an accompanying supplementary information file:

**S14. Fire activity**

[Figure]

**Figure S9:** 50-kyr records of total mercury (Hg$_T$) and mercury accumulation rate (Hg$_{AR}$) for Lake Bosumtwi from this study, with proxy datasets from prior studies of the same lake. These include forest (woody) taxa abundance represented by detrended correspondence analysis (DCA) axis 1 (Gosling et al., 2022a; Miller et al., 2016), percentage abundance of Poaceae (grass) pollen(Miller et al., 2016), microcharcoal concentrations(Miller et al., 2016), and macrocharcoal volume(Kiely, 2023). A distinct lake low stand (LS) based on seismic profiles and sedimentological data is marked between 33.5 and 32.8 m depth (grey shading)(Brooks et al., 2005; Scholz et al., 2007), and sapropel layer Unit S1 is marked between 3–5.5 m depth (brown shading)(Shanahan et al., 2012, 2006). Unit AI-1 is marked between 33.5 and 32.8 m depth (grey shading)(Brooks et al., 2005; Scholz et al., 2007), and sapropel layer Unit S1 is marked between 3–5.5 m depth (brown shading)(Shanahan et al., 2012, 2006).

Again, would it be worth doing some correlation analyses for data show in **Figure 5** to make your paper more quantitative?

> This is an interesting suggestion. On one hand, we do agree that there may be some value in performing analyses of this nature on the data presented in **Figure 5.** However, this would require (non-trivial) interpolation of nearly all datasets involved. Assessing whether false positive or negative results occur would be especially challenging given various differences in how age models were constructed and whether they fully account for any errors that may exist. Aside from the challenge of properly taking these chronological uncertainties into consideration, any correlations produced are in any case likely to highlight the correspondence (or lack thereof) with the most obvious changes in each record, and so would give limited additional insight beyond what is visible to the eye. Thus, we believe that the potential benefits of performing further statistical analyses on (interpolated versions of) the data presented in **Figure 5** do not sufficiently outweigh the risks of potential misinterpretation.

**Line 537**: seeks? How about this? This work combined geochemical data obtained for a sediment core collected from Lake Bosumtwi, Ghana, with climate data in order to explain measured trends in Hg concentrations.

> This statement does a great job at summarizing our study rationale. Inspired by this suggestion, we will amend the first line of our concluding section to read:

**Lines 592-594**: "*This study combines new sedimentary Hg data from Lake Bosumtwi, Ghana, with proxy data from archives across the African continent to explore whether hydroclimate has exerted a measurable effect on regional Hg cycling over the past ~96-kyr.*"

I like the schematic, **Figure 6**.

We are delighted that reviewer #2 found this figure a valuable addition to the manuscript, and are grateful for the positive feedback.
* * *
**We also thank Prof. Francus (editor) for taking the time to provide additional feedback on our work, and for the opportunity to further improve our manuscript. Please find below our detailed responses (in blue) to comments provided (repeated here in black).**

**Figure 3**: I understand that you have corrected the XRF output for the effect of compaction. However, XRF counts are compositional data that are summing at 100%. Therefore, there is a statistical bias because the data are not independent: for instance, if K increases, it means that other elements have to decrease. This can be accounted for by the Central Log Ratio (CLR) transformation. This is well described in Bertrand et al. (2024) Inorganic geochemistry of lake sediments: A review of analytical techniques and guidelines for data interpretation, Earth-Sci. Rev., 249, 10.1016/j.earscirev.2023.104639, 2024. I'm wondering if you could test whether the compositional nature of your XRF measurements is affecting the correlations presented in Figure 3.

This is an excellent point we had not previously considered, and agree is important to explore. Inspired by this suggestion, we proceeded to apply a CLR transformation to the XRF data used in our study, and subsequently re-visit the Pearson's correlation testing outlined in **section 3.6**. We also re-tested the significance of these correlations using the Student's t-test. Below, we compare correlation matrices based on analysis of the raw (a) and corrected (b) XRF data. We highlight that, although some slight changes are visible, the dominant trends remained the same: a negative overall correlation between $Hg_T$ and detrital materials (taken as $K_{clr}$ – r = -0.57; $r^2$ = 0.32), and between TOC and $K_{clr}$ (r = -0.59; $r^2$ = 0.34).

[Figure]

In light of this correction, we have made the following alterations to our manuscript:

- Added a description of the CLR approach to correction of XRF data in our methods:

**Lines 346 – 352**: "*To mitigate any effects arising from changes in the physical properties of the BOS04-5B sediments and/or the measurement times, we applied a centred-log ratio (clr) transformation to the measured XRF values. The centred log-ratio (clr) values are calculated by dividing the intensities of an element by the geometric mean of the intensities obtained on all selected elements, and are dimensionless such that positive values are generated for elements with high intensities, and vice versa (e.g., Bertrand et al. 2024). Therefore, elements (X) subject to this transformation are presented as their centred-log ratio value (Xclr).*"

- Replaced the correlation matrix previously presented in **Figure 3** (using raw XRF values – plot (a) above) to one in which analyses were run on the clr-corrected data (plot (b) above), and replaced original K (cps) with K$_{clr}$ in the lower panel (circled in orange below).
-

[Figure]

- Amended any statistical information, values, and results in the manuscript that have changed in response to the Pearson's re-testing (following CLR transformation). For example:

  **Lines 410 – 411**: "*The significance of all correlations was assessed using a Student's t-test, which showed that >50% of the assessed geochemical combinations were significant at p<0.01.*"

  **Lines 440 – 443**: "*However, it is noteworthy that detrital materials (e.g., K$_{clr}$) show negative correlations with both TOC (r = -0.59; r$^2$ = 0.34) and Hg (r = -0.57; r$^2$ = 0.32) so that the Hg-TOC correlation may reflect, in part, a correlation imposed by variable clay-dilution of both Hg and TOC.*"

  **Lines 454 – 455**: "*A negative overall correlation between Hg$_T$ and K$_{clr}$ is apparent throughout the record (r = -0.57; r$^2$ = 0.32; Fig. 3b).*"

  **Lines 468 – 469**: "*Strong correlations between Hg$_T$ and Mn$_{clr}$, and Hg$_T$ and Fe$_{clr}$ (redox sensitive elements) are also absent in BOS04-5B (Fig. 3).*"

- Added the clr-corrected XRF data to the supplementary data file that accompanies this manuscript.

- Amended supplementary figures **S4, S5,** and **S6** to display clr-corrected XRF, rather than the raw values as previously. Figure captions were also amended to include this statement:

  "*…X-ray fluorescence data measured using XRF by McKay (2012), and corrected using a centered-log ratio (clr) approach (Bertrand et al. 2024)*"

**Lines 416-419**: You write that there is no correlation between Mn and Fe (redox-sensitive elements) and Hg, suggesting that Hg concentration is not influenced by changes in redox conditions. This rationale is valid if all (or a great fraction of) Fe and Mn is affected by redox changes. However, this is seldom the case as, in common situations, a large part of Fe and Mn is locked in mineral phases that are insensitive to redox reactions. Do you have an idea of the mineral phases carrying Mn and Fe besides diagenetic carbonates? Maybe this can be tested by looking into the Mn/Ca or Fe/Ca along the sediment sequence?

> We agree this does warrant further clarification in the manuscript. Indeed, in normal oxic conditions Fe and Mn would be less relevant to consider with respect to Hg as both also have significant detrital and/or carbonate components that have no or an ambiguous relation with redox conditions. However, despite the ubiquitous occurrence of laminations throughout most of the Bosumtwi record which suggests near-permanent anoxic conditions, the occurrence of Mn-bearing siderite in the core, combined with the 'on-off' nature of the Mn signal, suggests irregular (subtle) changes in sediment pore water chemistry that are certainly worthy of closer assessment. Unlike Fe, the detrital fraction has a comparatively small contribution to Mn variability (see also figure included below), hence we find that Mn abundance provides the simplest path to assessing siderite abundance.

As the bottom waters in Lake Bosumtwi are considered near-permanently anoxic and given the ubiquitous presence of laminations, **we specifically consider the Mn signal. In this core, Mn dominantly reflects periods of siderite formation**, the clearest indicator of pore water redox changes. Although Mn is present in iron-sulfide minerals, vivianite, and ankerite, the intensity of the Mn peaks associated with these minerals is much lower (<100 cps) than with the manganosiderites (>1000 cps), and no other mineral has been identified in the sediments that corresponds to Mn intensities higher than 1000 cps (McKay, 2012).

Taking these factors into account, we now more clearly state in the manuscript that siderite specifically may play a role in sediment Hg cycling, as it may be involved in Hg reduction – hence our focus on the relation between the Hg and Mn (proxy for siderite concentration) signals.

> **Lines 469-472**: "*The majority of the examined record is marked by the presence of laminations, suggesting anoxic conditions dominate throughout (Shanahan et al. 2008). Although this means that redox changes were likely subtle, the coeval presence of siderite has potential implications for Hg through, for example, it's influence on Hg reduction (e.g., Ha et al. 2017).*"

> **Lines 475-482**: "*In this record we therefore test the relation between $Hg_T$ and Mn peaks. Pronounced Mn enrichments signal periods of (mangano)siderite formation, and this constitutes the clearest indicator of (subtle) pore water redox changes (McKay, 2012; Shanahan et al., 2008). Moreover, siderite specifically may be involved in Hg cycling through its potential to reduce Hg (e.g., Ha et al. 2017). Overall, the lack of evidence for substantial redox changes from ubiquitous laminations and the absence of a strong correlation between Mn and $Hg_T$ suggests that Hg concentrations in Lake Bosumtwi were not appreciably influenced by changes in redox conditions, nor the diagenetic effects signalled by these elements.*"

We have also sought to more explicitly state how Fe is present in the BOS04-5B core, and so have added the following statement to the manuscript:

> **Lines 472-475**: "*Fe is a major component of siderite, but it is also an important component of the detrital material that washes into Lake Bosumtwi and also potentially other redox-sensitive minerals precipitated in the lake and sediment pore waters (Shanahan 2006; Shanahan et al., 2008; 2009).*"

Regarding the examination of the Ca-normalized signals, we show Fe/Ca and Mn/Ca in BOS04-5B, alongside the $Hg_T$ data generated by this study (**Fig**. included below). We see that the only instance where these three parameters clearly co-vary is during sapropel 1 (S1). This is unsurprising given this sedimentary layer represents an extreme (end-member) example of anoxic conditions and likely coincides with a strong increase in Hg supply through precipitation (rain). However, we see no clear correspondence between Mn/Ca, Fe/Ca, and $Hg_T$ for the non-sapropelic intervals of our record, similar to the non-normalized Mn and $Hg_T$. We also make an interesting observation that Fe/Ca shows a similar pattern to both Mn and Mn/Ca when corrected for carbonate, which could highlight that diagenetic processes exert the dominant influence on Fe abundances in the BOS04-5B sediment; as opposed to detrital input. For example, if Ca reduction and Fe enrichment occur simultaneously.

[Figure]

**Lines 515-518**: You mention records from continental Africa, but **Figure 5** compares your record with marine records only, all of them being located at the west of Lake Bosumtwi, or in the front of the Nile delta. Is there any particular reason for that?

The decision to include these records in **Figure 5** was made in order to present an overview of West African climate since ~96 ka. Specifically, we wanted to include information on time resolved changes in African Monsoon strength (ODP 927), dust fluxes (ODP658, MD03-2705), Intertropical Convergence Zone position (MD03-2621), regional humidity (GeoB7920), and sea surface temperature (MD03-2707 – Gulf of Guinea). The records in **Figure 5** were selected on the basis their (a) proximity to Lake Bosumtwi, (b) resolution, and (c) temporal scope – three criteria which underpin why we present solely marine-based records in this figure.

Very few terrestrial paleoclimate records from West Africa are available, owing to the relative scarcity of depositional environments in this domain that can preserve long, continuous archives. As a result, terrestrial-based records remain underrepresented in West Africa relative to marine-based archives, and to terrestrial records sourced from the Eastern African Rift zone (e.g., Cohen et al. 2016). While these eastern-sourced records are important for discussion of broader trends in the paleoclimate of sub-Saharan Africa, they are also limited by their ability to describe the climate of West Africa due to an east–west trending moisture gradient, which leads to opposing dry and humid conditions between eastern and western Africa (e.g., Kaboth-Bahr et al. 2021). Therefore, we do not include them in **Figure 5**, and instead refer to several publications where these mechanisms are described in detail. For example:

> **Lines 580 – 589**: "*Paleoclimate records based on the sediments of lakes Malawi (Tanzania), Bambili (Cameroon), Tanganyika (Tanzania/Democratic Republic of the Congo), Chew Bahir (Ethiopia) and Chala (Tanzania) (e.g., **Cohen et al., 2007; Foerster et al., 2022; Lézine et al., 2019; Scholz et al., 2007**), and marine sediment core material from the West African margin (Figs. 1, 5) (e.g., Kinsley et al., 2022; Skonieczny et al., 2019) also pertain to a distinct regional-scale shift in hydro climate across tropical sub-Saharan Africa at this point in time (Fig. 5). Specifically, a shift characterized by a distinct moisture gradient favouring wetter conditions in the west of the continent relative to the east, which was further amplified during the last glacial termination (e.g., **Baxter et al., 2023; Gosling et al., 2022a; Lupien et al., 2023**).*"